

**Quantifying Soil Carbon Accumulation in Alaskan Terrestrial Ecosystems during the Last**
**15,000 Years**
Sirui Wang[1], Qianlai Zhuang[1,2*], Zicheng Yu[3]
[1]Department of Earth, Atmospheric, and Planetary Sciences, Purdue University, West Lafayette,
Indiana, 47907
[2]Department of Agronomy, Purdue University, West Lafayette, IN 47907
[3]Department of Earth and Environmental Sciences, Lehigh University, Bethlehem, PA 18015
Correspondence to: qzhuang@purdue.edu




**Abstract:** Northern high latitudes contain large amounts of soil organic carbon (SOC), in which Alaskan terrestrial ecosystems account for a substantial proportion. In this study, the SOC accumulation in Alaskan terrestrial ecosystems over the last 15,000 years was simulated using a process-based biogeochemistry model for both peatland and non-peatland terrestrial ecosystems. Comparable with the previous estimates of 25-70 Pg C in peatland and 13-22 Pg C in non-peatland soils within 1-m depth in Alaska, our model estimated a total SOC of 36-63 Pg C at present, including 27-48 Pg C in peatland soils and 9-15 Pg C in non-peatland soils. Vegetation stored only 2.5-3.7 Pg C in Alaska currently with 0.3-0.6 Pg C in peatlands and 2.2-3.1 Pg C in non-peatlands. The simulated average rate of peat C sequestration was 2.3 Tg C yr$^{-1}$ with a peak value of 5.1 Tg C yr$^{-1}$ during the Holocene Thermal Maximum (HTM) in the early Holocene, four folds higher than the average rate of 1.4 Tg C yr$^{-1}$ over the rest of the Holocene. The SOC accumulation slowed down, or even ceased, during the neoglacial climate cooling after the mid-Holocene, but accumulation increased again in the 20th century. The model-estimated peat depths ranged from 1.1 to 2.7 m, similar to the field-based estimate of 2.29 m for the region. We found that the changes in vegetation types and their distributions due to climate change were the main factors determining the spatial variations of SOC accumulation during different time periods. Warmer summer temperature and stronger radiation seasonality, along with higher precipitation in the HTM and the 20th century might have resulted in the extensive peatland expansion and carbon accumulation, implying that soil C accumulation would continue under future warming conditions.

**Keywords:** Carbon, Peatlands, Alaska, Modelling, Climate



## 1. Introduction

Global surface air temperature has been increasing since the middle of the 19[th] century (Jones and Mogberg, 2003; Manabe and Wetherald, 1980, 1986). Since 1970, the warming trend has accelerated at a rate of 0.35 ℃ per decade in northern high latitudes (Euskirchen et al., 2007; McGuire et al., 2009). It is predicted that the warming will continue in the next 100 years (Arctic Climate Impact Assessment 2005; Intergovernmental Panel on Climate Change (IPCC), 2013, 2014). The land surface in northern high latitudes (>45° N) occupies 22% of the global surface and stores over 40% of the global soil organic carbon (SOC) (McGuire et al., 1995; Melillo et al., 1995; McGuire and Hobbie, 1997). Specifically, the northern high latitudes were estimated to store 200-600 Pg C (1 Pg C = $10^{15}$ g C) in peatland soils depending on the depth considered (Gorham, 1990, 1991; Yu, 2012), 750 Pg C in non-peatland soils (within 3 m) (Schuur et al., 2008; Tarnocai et al., 2009; Hugelius et al., 2014), and additional 400 Pg C in frozen loess deposits of Siberia (Zimov et al., 2006a). Peatland area is around 40 million hectares in Alaska compared with total 350 million hectares in northern high-latitude regions (Kivinen and Pakarinen, 1981). Alaskan peatlands account for the most vast peatland area in the USA and cover at least 8% of total land area (Bridgham et al., 2006). To date, the regional soil C and its responses to the climate change are still with large uncertainty (McGuire et al., 2009; Loisel et al., 2014).

The warming climate could increase C input to soils as litters through stimulating plant net primary production (NPP) (Loisel et al., 2012). However, it can also decrease the SOC by increasing soil respiration (Yu et al., 2009). Warming can also draw down the water table in peatlands by increasing evapotranspiration, resulting in a higher decomposition rate as the aerobic respiration has a higher rate than anaerobic respiration in general (Hobbie et al., 2000).



SOC accumulates where the rate of soil C input is higher than decomposition. The variation of
climate may switch the role of soils between a C sink and a C source (Davidson and Janssens,
2006; Davidson et al., 2000; Jobbagy and Jackson, 2000). Unfortunately, due to the data gaps of
field-measurement and uncertainties in estimating regional C stock (Yu, 2012), with limited
understanding of both peatlands and non-peatlands and their responses to climate change, there is
no consensus on the sink and source activities of these ecosystems (Frolking et al., 2011; Belyea,
2009; McGuire et al., 2009).
To date, both observation and model simulation studies have been applied to understand
the long-term peat C accumulation in northern high latitudes. Most field estimations are based on
series of peat-core samples (Turunen et al., 2002; Roulet et al., 2007; Yu et al., 2009; Tarnocai et
al., 2009). However, those core analyses may not be adequate for estimating the regional C
accumulation due to their limited spatial coverage. Model simulations have also been carried out.
For instance, Frolking et al. (2010) developed a peatland model considering the effects of plant
community, hydrological dynamics and peat properties on SOC accumulation. The simulated
results were compared with peat-core data. They further analyzed the contributions of different
plant functional types (PFTs) to the peat C accumulation. However, this 1-D model has not been
used in large spatial-scale simulations by considering other environmental factors (e.g.,
temperature, vapor pressure, and radiation). In contrast, Spahni et al. (2013) used a dynamic
global vegetation and land surface process model (LPX), based on LPJ (Sitch et al., 2003),
imbedded with a peatland module, which considered the nitrogen feedback on plant productivity
(Xu-Ri and Prentice, 2008) and plant biogeography, to simulate the SOC accumulation rates of
northern peatlands. However, these models have not been evaluated with respect to their
simulations of soil moisture, water table depth, methane fluxes, and carbon fluxes presumably



due to relatively simple model structures, especially in terms of ecosystem processes (Stocker et
al., 2011, 2014; Kleinen et al., 2010). Furthermore, climatic effects on SOC were not fully
explained. The Terrestrial Ecosystem Model (TEM) has been applied to study C and nitrogen
pools and fluxes in the Arctic (Zhuang et al., 2001, 2002, 2003, 2015; He et al., 2014). However,
the model has not been calibrated and evaluated with peat-core C data, and has not been applied
to investigate the peatland C dynamics. Building upon these efforts, recently we fully evaluated
the peatland version of TEM (P-TEM) including modules of hydrology (HM), soil thermal
(STM), C and nitrogen dynamics (CNDM) for both upland and peatland ecosystems (Wang et
al., 2016).

Here we used the peatland-core data for various peatland ecosystems to parameterize and

test P-TEM (Figure 1). The model was then used to quantify soil C accumulation of both
peatland and non-peatland ecosystems across the Alaskan landscape since the last deglaciation.
This study is among the first to examine the current peatlands and non-peatlands C distributions
and peat depths in various ecosystems at the regional scale.

**2. Methods**
**2.1 Model Description**

In P-TEM, peatland soil organic C (SOC) accumulation is determined by the difference

between the net primary production (NPP) and aerobic and anaerobic decompostion. Peatlands
accumulate C where NPP is greater than decomposition, resulting in positive net ecosystem
production (NEP):



$$NEP = NPP - R_H - R_{CH_4} - R_{CWM} - R_{CM} - R_{COM} \quad (1)$$

P-TEM was develoepd based on the Terrestrial Ecosystem Model (TEM) at a monthly

step (Zhuang et al., 2003; 2015). It explicitly considers the process of aerobic decomposition
($R_H$) related to the variability of water-table depth; net methane emission after methane oxidation
($R_{CH_4}$); $CO_2$ emission due to methane oxidation ($R_{CWM}$) (Zhuang et al., 2015); $CO_2$ release
accompanied with the methanogenesis ($R_{CM}$) (Tang et al., 2010; Conrad, 1999); and $CO_2$ release
from other anaerobic processes ($R_{COM}$, e.g., fermentation, terminal electron acceptor (TEA)
reduction) (Keller and Bridgham, 2007; Keller and Takagi, 2013). For upland soils, we only
considered the heterotrohic respiration under aerobic condition (Raich, 1991). For detailed model
description see Supplement.

We model peatland soils as a two-layer system for hydrological module (HM) while

keeping the three-layer system for upland soils (Zhuang et al., 2002). The soil layers above the
lowest water table position are divided into: (1) moss (or litter) organic layer (0-10 cm); and (2)
humic organic layer (10-30 cm) (Wang et al., 2016). Based on the total amount of water content
within those two unsaturated layers, the actual water table depth ($WTD$) is estimated. The water
content at each 1 cm above the water table can be then determined after solving the water
balance equations (Zhuang et al., 2004).

In the STM module, the soil vertical profile is divided into four layers: (1) snowpack in

winter, (2) moss (or litter) organic layer, (3) upper and (4) lower humic organic soil (Wang et al.,
2016). Each of these soil layers is characterized with a distinct soil thermal conductivity and heat
capacity. We used the observed water contents at the particular sites to drive the STM (Zhuang et
al., 2001).



The methane dynamics module (MDM) (Zhuang et al., 2004) considers the processes of

methanogenesis, methanotrophy, and the transportation pathways including: (1) diffusion
through the soil profile; (2) plant-aided transportation; and (3) ebullition. The soil temperatures
calculated from STM, after interpolation into 1-cm sub-layers, are input to the MDM. The water
table depth and soil water content in the unsaturated zone for methane production and emission
are obtained from HM, and the net primary production (NPP) is calculated from the CNDM.
Soil-water pH is prescribed from observed data and the root distribution determines the redox
potential (Zhuang et al., 2004).
**2.2 Model Parameterization**

We have parameterized the key parameters of the individual modules including HM,

STM, and MDM (Wang et al., 2016). The parameters in CNDM for upland soils and vegetation
have been optimized in the previous studies (Zhuang et al 2002, 2003; Tang and Zhuang 2008).
The parameters for peatland soils in P-TEM were parameterized using a moderate rich
*Sphagnum* spp. open fen (APEXCON) and a *Sphagnum*-black spruce (*Picea mariana*) bog
(APEXPER) (Table 3). Both are located in the Alaskan Peatland Experiment site (APEX) study
area, where *Picea mariana* is the only tree species above breast height in APEXPER. Three
water table position manipulations were established in APEX including a control, a lowered, and
a raised water table plots (Chivers et al., 2009; Turetsky et al., 2008; Kane et al., 2010; Churchill
et al., 2011). There were also several internal collapse scars that formed with thaw of surface
permafrost, including a non-, an old, and a new collapse plots. APEXCON represents the control
manipulation and APEXPER represents the non-collapse plot. The annual NPP and aboveground
biomass at both sites have been measured in 2009. There were no belowground observations;
however, at a Canadian peatland, Mer Bleue, which includes *Sphagnum* spp. dominated bog





(dominated by shrubs and *Sphagnum*) and pool fen (dominated by sedges and herbs and
*Sphagnum*). Assuming the belowground biomass in APEXCON and APEXPER was close to that
in Mer Bleue, we used the belowground biomass at Mer Bleue to represent the missing
observations at both sites (Table 4). We conducted a set of 100,000 Monte Carlo ensemble
simulations for each site-level calibration, and parameters with the highest mode in posterior
distribution were selected (Tang and Zhuang, 2008, 2009).
**2.3 Regional Vegetation Data**
The Alaskan C stock was simulated through the Holocene where the vegetation biome
maps were reconstructed at four time periods: a time period encompassing a millennial-scale
warming event during the last deglaciation known as the BØlling-AllerØd at 15-11 ka (1 ka =
1000 cal yr Before Present), HTM during the early Holocene at 11-10 and 10-9 ka as well as the
mid- (9-5 ka) and late- Holocene (9 ka-1900 AD) (He et al., 2014).  We used the modern
vegetation distribution for the simulation during the period 1900-2000 AD (Figure 2). We
assumed that the vegetation distribution remained static within each corresponding time period.
Five vegetation types were classified as upland vegetation: boreal deciduous broadleaf forest,
boreal evergreen needleleaf and mixed forest, alpine tundra, wet tundra; and barren (Table 1).
Mountain ranges and large water bodies were delineated as 'Barren' and data could not be
interpolated across them. By using the same vegetation distribution map, we reclassified the
upland vegetation into two peatland vegetation types: *Sphagnum* spp. poor fens (SP) generated
from tundra ecosystems, and *Sphagnum* spp-black spruce (*Picea mariana*) bog/ peatland (SBP)
generated from forest ecosystems (Table 1), both of which dominate the major area of Alaskan
peatlands. We used both the upland and peatland vegetation types to simulate the C dynamics in
Alaska.



Upland and peatland distribution for each grid cell was determined using the wetland
inundation data extracted from the NASA/ GISS global natural wetland dataset (Matthews and
Fung, 1987). The resolution was resampled to $0.5° \times 0.5°$ from $1° \times 1°$. We postulated that,
given the same topography of Alaska during the Holocene, it was reasonable to assume that the
wetland distribution can be represented by modern inundation map. The inundation fraction was
assumed to be the same within each grid through time and the land grids not covered by
expanded peatland yet were assumed as uplands. We calculated the total area of modern Alaskan
peatlands to be 302,410 $km^2$, which was within the range from 132,000 $km^2$ (Bridgham et al.,
2006) to 596,000 $km^2$ (Kivinen and Pakarinen, 1991). The soil water pH data were extracted
from Carter and Scholes (2000), and the elevation data were derived from Shuttle Radar
Topography Mission and were resampled to $0.5° \times 0.5°$ spatial resolution.
**2.4 Climate Data**
Climate data were downscaled and bias-corrected from ECBilt-CLIO model output
(Timm and Timmermann, 2007; He et al., 2014). Climate fields include monthly precipitation,
monthly air temperature, monthly net incoming solar radiation, and monthly vapor pressure
$(2.5° \times 2.5°)$. We used the same time-dependent forcing atmospheric carbon dioxide
concentration data for model input as were used in ECBilt-CLIO transient simulations from the
Taylor Dome (Timm and Timmermann, 2007). The historical climate data used for the
simulation through the 20[th] century are monthly CRU2.0 data.
The mean annual net incoming solar radiation (NIRR) was $78\pm4.8$ W $m^{-2}$ before the
HTM (15-11 ka). It showed an increase at the early HTM (11-10 ka), reaching $83.6\pm4.5$ W $m^{-2}$
and continueed to increase to $84\pm4.7$ W $m^{-2}$ at the late HTM (10-9 ka). NIRR decreased after



the HTM through the entire mid-Holocene (9-5 ka) to a minimum of 79$\pm$5 W m$^{-2}$ at the end of
the Holocene. It became higher from 1900 to 2000 AD, with annual mean 82$\pm$5.1 W m$^{-2}$
(Figure 3b). The mean annual air temperature showed a similar pattern as it rose from -7$\pm$1.8 ℃
to -5$\pm$1.6 ℃ at the early HTM and reached -4.7$\pm$1.5 ℃ at the late HTM, indicating a warmer-
than-present climate. There was also a temperature decrease when HTM ended through the rest
of the Holocene and the temperature increased again from 1900 AD to -5.8$\pm$1.5 ℃, presumably
due to the global warming (Figure 3d). Total annual precipitation increased from 306$\pm$40 mm to
369$\pm$25 mm at the end of the HTM, suggesting an overall wet climate. A dryer condition
occurred from the mid-Holocene and became driest in the late-Holocene (5 ka-1900 AD) (Figure
3f). The monthly values of NIRR followed the same pattern as annual means, except during the
winter. The maximum summer radiation occurred during the late-HTM, leading to the highest
radiation seasonality. Large seasonality also appeared in the 20[th] century, however, lower than
that during the HTM (Figure 3a). Temperature seasonality followed the trend of annual
temperature. The days of year with temperature above 0 ℃ increased 10-15 days at the HTM
compared with that before the HTM, suggesting a longer growing season (Figure 3c).
Precipitations were highest during the summer (July-September) in each time period and lowest
during the winter and early spring (December-April). The periods at 15-11ka and in the late-
Holocene exhibited less overall, especially summer precipitations than at the HTM. During the
20[th] century, there was less winter precipitation but it was compensated by a higher summer
precipitation compared with the late-Holocene (Figure 3e). The orbital induced maximum
seasonality of insolation and the warmest climate during the HTM as described in Huybers et al.
(2006) and Yu et al. (2010) corresponded well to the simulated trends of air temperature.





**2.5 Data of Peatland Basal Ages**

We conducted the simulation from 15 to 5 ka for an Alaskan peatland assuming it started

to accumulate C since 15 ka. However, assuming that peatlands in all grids had the same basal
age (15 ka) could overestimate the total peat SOC accumulation. Therefore we used the observed
basal ages of peat samples from Gorham et al. (2012) and categorized them into different time
periods (Figure 2). We found that during each period, the spatial distribution of peatland basal
ages was similar to that of the vegetation types (e.g., peatland initiation points were mainly
located where was dominated by alpine tundra at south, northwestern, and southeastern coast
during 15-11 ka). We thus used the vegetation types to estimate the peatland basal ages at
regional scales (Table 2).
**2.6 Simulations and Sensitivity Test**

To verify the model ability to simulate the peat C accumulation rates in the past 15,000

years, we conducted a simulation using pixels located on the Kenai Peninsula from 15 to 5 ka
after model parameterization. We compared the model simulation results with the peat-core data
from four peatlands on the Kenai Peninsula, Alaska (Jones and Yu, 2010; Yu et al., 2010) (see
Wang et al. (2016) for detail).  The observed data include the peat depth, bulk density of both
organic and inorganic matters at 1-cm interval, and age determinations. The simulated C
accumulation rates represent the actual ("true") rates at different times in the past. However, the
calculated accumulation rates from peat cores are considered as "apparent" accumulation rates,
as peat would continue to decompose since the time of formation until present when the
measurement was made (Yu, 2012). To facilitate comparison between simulated and observed
accumulation rates, we converted the simulated "true" accumulation rates to "apparent" rates,



following the approach by Spahni et al. (2013). That is, we summed the annual net C
accumulation over each 500-year interval and deducted the total amount of C decomposition
from that time period, then dividing by 500 years.

For the study region, we conducted a transient simulation using continuous monthly

meteorology data (Figure 2) from 15 ka to 2000 AD. Five maps (Figure 3) were used to represent
the vegetation distributions of Alaska and were assumed to be static during each time period
(e.g., 15-11 ka, 11-10 ka, 10-9 ka, 9 ka-1900 AD, and 1900-2000 AD). The simulation was
firstly conducted assuming all grid cells were taken up by upland vegetation to get the upland
soil C spatial distributions during different time periods. We then conducted the second
simulation assuming all grid cells were dominated by peatland vegetation by merging upland
types into peatland types following Table 1 to obtain the distributions of peat SOC accumulation.
We used the inundation fraction map to extract both uplands and peatlands from each grid and
estimated the corresponding SOC stocks within each grid, which were then summed up to
represent the Alaskan SOC stock.

We conducted a sensitivity test to evaluate the responses of NPP, SOC decomposition

rates (aerobic plus anaerobic respiration), and net SOC balance to the climate variables.
Simulations under three scenarios were conducted to test the temperature effect. We used the
original forcing data as the standard scenario and the warmer (monthly temperature +5℃) and
cooler (−5℃) as other two while keeping the rest forcing data unchanged. Similarly, we used the
original forcing data as the standard scenario and the wetter (monthly precipitation +10 mm) and
drier (−10 mm) to test the effect from precipitation. To further study if vegetation distribution
has stronger effects on SOC sequestration than climate in Alaska, we simply replaced SBP with
SP and simultaneously replaced the upland forests with tundra at the beginning of 15 ka. We also



conducted the simulation under "warmer" and "wetter" conditions described before while
keeping the vegetation distribution unchanged.

## 3. Results

### 3.1 Simulated Peatland Carbon Accumulation Rates at Site Level

Our paleo simulation showed a large peak of peat C accumulation rates at 11-9 ka during
the HTM (Figure 4). The simulated "true" and "apparent" rates captured this primary feature in
peat-core data at almost all sites (Jones and Yu, 2010). The simulated magnitude of this peak was
similar to observations at No Name Creek and Horse Trail Fen, but overestimated at Kenai
Gasfield and Swanson Fen at 10-9 ka (late-HTM). The secondary peak of C accumulation rates
appeared at 6-5 ka in the mid-Holocene. The simulation successfully estimated both peaks at
Swanson Fen, No Name Creek, and Kenai Gasfield, but with overestimated magnitude at
Swanson Fen. The comparison between simulation and observation using averages in 500-year
bins revealed a high correlation ($R^2 = 0.90, 0.88,$ and $0.39$), especially at No Name Creek and
Horse Trail Fen. The simulated SOC accumulation rates corresponded well to the synthesis
curves at four sites (Figure 4b).

### 3.2 Vegetation Carbon Storage

Model simulations showed an overall low mean annual vegetation C storage before the
HTM (15-11 ka) (Figure 5a), paralleled to the relatively low annual NPP (Figure 5b). The
*Sphagnum*-dominated peatland represented the lowest vegetation C storage (2.5 kg C m$^{-2}$),
much lower than the *Sphagnum*-black spruce peatland (1 kg C m$^{-2}$). Upland vegetation showed a
generally higher C storage, with the highest amount of C stored in boreal evergreen needleleaf
forests (2 kg C m$^{-2}$). The upland forests also showed a higher rate of annual NPP (0.31-0.35



kg C m$^{-2}$yr$^{-1}$). C storage of alpine and moist tundra was higher than peatlands, while the annual
NPP were lower (0.08-0.1 kg C m$^{-2}$yr$^{-1}$). Higher NPP were shown in almost all vegetation
types during the early Holocene. There were no significant changes of vegetation C storage in
peatlands and tundra compared with boreal forests. All vegetation showed a higher NPP and
vegetation C during the late-HTM. Mean annual vegetation C exceeded 0.5 g C m$^{-2}$ and 1.3
g C m$^{-2}$ for *Sphagnum* and black spruce peatlands. Evergreen forest stored over 4.7 kg C m$^{-2}$.
During the mid-Holocene, almost all vegetation types represented a decrease in both NPP and
vegetation C. The plant productivity along with the vegetation C began to slightly increase at
late-Holocene and became stable, possibly resulted from the rising temperature.

Approximately 2 Pg C was stored in both upland and peatland vegetation in Alaska

before the HTM (Figure 6). Upland moist tundra accounted for the most amount of C due to its
large area. At the early HTM, evergreen needleleaf forest area became the largest, and about 1.9
Pg C was stored in boreal forests. More C was stored in black spruce peatland also because of
the forest formation. Boreal forest accounted for 3.5 Pg C at the late HTM. Decrease of
vegetation C occurred at mid-Holocene. The simulation through the Holocene to present
indicated that the lowest amount C was stored in vegetation before the HTM, while vegetation
assimilated the largest amount of C during the late-Holocene. We estimated a total 2.9 Pg C
stored in modern Alaskan vegetation, with 0.4 Pg in peatlands and 2.5 Pg in non-peatlands. The
uncertainties of the parameters during the model calibration (Table 4) resulted in a range of 0.3-
0.6 Pg C and 2.2-3.1 Pg C in peatlands and non-peatlands, respectively.




### 3.3 Soil Carbon Stocks

Carbon storage in Alaskan non-peatland soils varied spatially (Figure 7). Generally, deciduous broadleaf forests had a higher SOC (8-13 kg C m$^{-2}$) than evergreen needleleaf forests (3-8 kg C m$^{-2}$), while moist tundra had the highest SOC (12-25 kg C m$^{-2}$). The SOC showed an overall increase in both boreal forests and moist tundra during the early-HTM (11-10 ka) (Figures 7a, b). With the continued expansion of the boreal forests during the late-HTM (10-9 ka) (Figure 4c), the spots of low SOC concentration were widely spread (Figure 7c). During the mid- (9-5 ka) and late-Holocene (5 ka-1900 AD), although the wet tundra took back the most area, the SOC decreased (Figure 7d) presumably due to the cooler and drier conditions, which was consistent with the decline in mean annual NPP and vegetation C (Figure 5). An increase occurred again in the last century with mean SOC comparable to the late-HTM (Figure 7f). An average of 3.1 Pg C was simulated before the HTM (Figure 8). The SOC increased sharply during the early-HTM (to 11.5 Pg C) across Alaska and slightly decreased to 9 Pg C at the end of HTM. There was little variation during the mid- and late-Holocene (10.7 Pg C) and the amount increased to 11.2 Pg C at the end of the 20[th] century. Due to model parameterization (Table 4), the regional soil C estimates ranged from 9 to 15 Pg C at present.

The peatland SOC showed a different pattern compared to upland soils. Peatlands started to accumulate C at 15 ka mainly in northwestern, southeastern, and south coastal regions of Alaska (Figure 9a). Much less C (<10 kg C m$^{-2}$) was accumulated in the southeastern coast in comparison to other coastal parts (>15 kg C m$^{-2}$). Initially, only *Sphagnum* open peatland (SP) existed, with no *Sphagnum*-black spruce forested peatland (SBP). At the beginning of the HTM, there was a peatland area of ~4.5× $10^5$ km² (Figure 10). During the early-HTM, the SP formed in the north coast and the SBP rapidly expanded in south coast and east central regions,



becoming the dominant peatland type in Alaska (Figure 9b). Meanwhile the peatlands area
increased to ~13× $10^5$ km$^2$ (Figure 10). The SBP continued to expand to the central Alaska
during the late-HTM (Figure 9c). Although peatlands continued to form towards west in the mid-
Holocene (Figures 9d, 10), some areas that were dominated by SBP in interior Alaska stopped
accumulating SOC. By the end of the mid-Holocene, almost all the peatlands have formed
(Figure 10) and some grids showed negative accumulation in the late-Holocene (Figure 9e).
However, as the global warming began in the 20$^{th}$ century, SOC accumulation increased rapidly
again (Figure 9f).

The mean annual SOC accumulation rates increased from 0.9 to 28.7 g C m$^{-2}$yr$^{-1}$and

from 0 to 57.1 g C m$^{-2}$yr$^{-1}$ in the early-HTM (11-10 ka) for SP and SBP, respectively, with an
area-weighted rate of 41.6 C m$^{-2}$yr$^{-1}$ (Figure 11). The accumulation rate of the SP increased to
48.6 g C m$^{-2}$yr$^{-1}$ while the rate of SBP slightly decreased to 56.7 g C m$^{-2}$yr$^{-1}$ with an overall
rate 54.7 C m$^{-2}$yr$^{-1}$ in the late-HTM (10-9 ka) (Figure 11), followed by a drop to 22.7 and 13.1
g C m$^{-2}$yr$^{-1}$ in the mid-Holocene (Figure 11). Late-Holocene rates ranged from 9.8 to -8.0
g C m$^{-2}$yr$^{-1}$ for SP and SBP. The rates of SP and SBP reached 42.5 and 33.2 g C m$^{-2}$yr$^{-1}$
respectively in the 20$^{th}$ century.

The change in total SOC stock corresponded well to the mean annual accumulation rates

during the last 15,000 years (Figures 8, 11). A total of 37.4 Pg C was estimated to accumulate in
Alaskan peatlands, with 23.9 Pg C in SP and 13.5 Pg C in SBP, from 15 ka to 2000 AD. The
total peat C stock had an uncertainty range of 27-48 Pg C depending on model parameters (Table
4). The peatlands in the northern and southern coastal regions showed the highest SOC densities
(>150 kg C m$^{-2}$), while some central regions had the lowest (<20 kg C m$^{-2}$) (Figure 12a). For
newly formed peatlands in west central part and west coast, <100 kg C m$^{-2}$ SOC was



accumulated. The non-peatland SOC distribution was mainly decided by the vegetation types,
with high densities (>15 kg C m$^{-2}$) in west and north coast where tundra dominated and low
densities (<10 kg C m$^{-2}$) in central and east parts where boreal forests dominated (Figure 12b).

We used the observed mean C content of 46.8% in peat mass and bulk density of 166±76

kg m$^{-3}$ in Alaska (Loisel et al., 2014) to estimate peat depth at each peat grid cell from the
simulated peat SOC density (kg C m$^{-2}$). The spatial pattern of peat depth is identical to the SOC
distribution, with most regions having peat depths of <2.5 m (Figure 12c). Based on the modern
land area in each TEM gird cell and the inundation map, we estimated a weighted average depth
of 1.9 m (ranging from 1.1 to 2.7 m, considering uncertainty in bulk density values) for Alaska
peatlands. We also combined the SOC in both peatlands and non-peatlands results together to
generate the total SOC distribution (Figure 12d). Soils at northern coast had the highest densities,
many grids had SOC >40 kg C m$^{-2}$. Southwestern coast and eastern central Alaska also showed
a high total SOC accumulation (>40 kg C m$^{-2}$). Central, eastern parts and west coast had the
lowest SOC densities (<20 kg C m$^{-2}$).
**3.4 Sensitivity Test**

We found that NPP and decomposition rates changed simultaneously, but NPP had the

dominant effect as the net SOC accumulation rate of Alaska increased and decreased under
warmer and cooler conditions, respectively (Figures 13a, c, e). The net SOC accumulation rate
increased as the condition became wetter and vice versa (Figures 13b, d, f). We also found an
increase of SOC from 11.2 to 14.6 Pg C for upland mineral soils and 37.5 to 71 Pg C for
peatlands after replacing the SP to SBP and upland forest systems with tundra. Meanwhile, under



"warmer" and "wetter" conditions, the upland and peatland SOC increased by 13.8 Pg C and 35
Pg C, respectively.
**4. Discussion**
**4.1 Effects of Climate on Ecosystem Carbon Accumulation**

The simulated climate by ECBilt-CLIO model showed that among the six time periods, the

coolest temperature appeared at 15-11 ka, followed by the late Holocene (5 ka-1900 AD). Those
two periods were also generally dry (Figure 3f). The former represented colder and drier climate
before the onset of the Holocene and the HTM (Barber and Finney, 2000; Edwards et al., 2001).
The latter represented post-HTM neoglacial cooling, which caused permafrost aggradation
across northern high latitudes (Oksanen et al., 2001; Zoltai, 1995).

The simulated NPP, vegetation C density and storage were highest during the HTM

(Figures 5, 6). The highest C accumulation rates in both peatlands and non-peatlands occurred at
the time (Figures 7-11). ECBilt-CLIO simulated an increase of temperature in the growing
season (Figure 3c), also leading to a stronger seasonality of temperature during the HTM
(Kaufman et al., 2004, 2016), caused by the maximum summer insolation (Berger and Loutre,
1991; Renssen et al., 2009). The highest mean annual and highest summer precipitations were
also simulated during the 10-9 ka period. The highest vegetation C uptake and SOC
accumulation rates coincided with the warmest summer and the wetter-than-before conditions,
suggesting a strong link between those climate variables and C dynamics in Alaska. Enhanced
climate seasonality characterized by warmer summer, enhanced summer precipitation and
possibly earlier snow melt during the HTM increased NPP, as shown in our sensitivity test.
Annual NPP increased by 40 and 20 g C m$^{-2}$ yr$^{-1}$ under the warmer and wetter scenarios,



respectively (Figures 13a, b), indicating summer temperature and precipitation were the primary
controls over NPP. Warmer condition could positively affect the SOC decomposition (Nobrega
et al., 2007). Furthermore, hydrological effect can also be significant as higher precipitation
could raise the water-table position, allowing less space for aerobic respiration. As shown in the
sensitivity test, warmer and wetter could lead to an increase of decomposition up to 35 and 15
$g\ C\ m^{-2}\ yr^{-1}$, respectively (Figures 13c, d). Such climatic effects on ecosystem productivity
were consistent with modern studies (Tucker et al., 2001; Kimball et al., 2004; Linderholm,
2006). Our results did not show a decrease in total heterotrophic respiration throughout Alaska
from the higher precipitation, presumably due to a much larger area of upland soils ($1.3 \times 10^6$
$km^2$) than peatland soils ($0.26 \times 10^6\ km^2$), as higher precipitation would cause higher aerobic
respiration in the unsaturated zone of upland soils. The relatively low vegetation NPP and C
density, along with the low total vegetation and soil C stocks during 15-11 ka period and late-
Holocene were consistent with the unfavorable cool and dry climate conditions (Figures 5, 6, 8,
11). Our previous simulations at four peatland sites in Alaska (Wang et al., 2016) suggested that
temperature had the most significant effect on peat accumulation rate, followed by the
seasonality of net solar radiation and temperature. Precipitation and the interactive effect from
temperature and precipitation had some certain effects ($p<0.05$). The period from 15 to 11 ka
experienced lower snowfall than the HTM. The combination of decreased snowfall and lower
temperature could result in deeper frost depth due to the decreased insulative effects of the
snowpack, and therefore shortening the period for active photosynthetic C uptake, leading to an
overall low productivity (McGuire et al., 2000; Stieglitz et al., 2003). The positive effect of
temperature on SOC accumulation as shown in this study, may help explain the coincidence
between low SOC accumulation rates across the northern peatland domain and the cooler





condition during the neoglacial period (Marcott et al., 2013; Vitt et al., 2000; Peteet et al., 1998;
Yu et al. 2010). The stimulation of SOC accumulation from the warming and the rapid SOC
accumulation rates during the 20th century in our study suggested a continue C sink will exist
under the warmer and wetter climate conditions in the 21st century, as also concluded in Spahni
et al. (2013).

The 20th century represented a temperature rise induced by global warming. It was still

1.1 ℃ lower than the late-HTM, suggesting the warmest climate during the HTM, which agreed
with the previous study (Stafford et al., 2000). It was also lower than the mid-Holocene, which
compared favorably with other estimates (Anderson and Brubaker, 1993; Kaufman et al., 2004).
However, the annual precipitation during modern time estimated from other studies was higher
than the HTM and mid-Holocene (Barber and Finney, 2000). The model output we used may
overestimate the precipitation in the HTM, which could subsequently overestimate the water-
table position and thus, the annual C accumulation rates. As studied, regional precipitation varies
largely depending on the local topography (Stafford et al., 2000), thus the estimates with large-
scale climate models have a large uncertainty. Great heterogeneity is produced from using large
climatic controls (e.g., insolation and sea ice extent), which casts limits for accurately simulating
the location- and topographic-specific climate data, especially precipitation (Whitlock and
Bartlein, 1993; Mock and Bartlein, 1995).

**4.2 Effects of Vegetation Distribution on Ecosystem Carbon Accumulation**

Different vegetation distributions during various periods led to clear step changes,

suggesting vegetation composition is likely to be the primary control on C dynamics. Similarly,
SBP areas stored lower C than SP in overall at the spatial scale during each time period (Figure



9). Under cooler and drier climates, forested peatlands generally stopped accumulating SOC
during the mid- and late-Holocene with some areas accompanied by a negative accumulation rate
(Figures 9d,e), suggesting that such type of peatland could be more vulnerable to climate change
due to its low C storage.
As key parameters controlling C dynamics in the model (e.g., maximum rate of
photosynthesis, litter fall C) are ecosystem type specific, vegetation distribution change may
have a dominant effect on simulated regional plant productivity and C storage. Our sensitivity
test indicated that by replacing all vegetation types with forest systems, there was a total increase
of 36.9 Pg in upland and peatland soils. There was also an increase of 48.8 Pg C under warmer
and wetter conditions. These tests indicated that both climate and vegetation distribution have
significant effects on C storage.
However, the high correlation between climate and ecosystem C dynamics as discussed
above indicated that climate was probably the fundamental driver for vegetation composition
changes over time. The vegetation changes as reconstructed from fossil pollen data during
different time periods followed the general climate history during the last 15,000 years (He et al.,
2014). Upland alpine and moist tundra stored the largest amounts of C due to their large areas
among all vegetation types, as forests areas were limited before the HTM (Figure 6). On the
basis of the observed relationship between the distributions of basal ages of peat samples and
vegetation types (Table 2, Figure 2), alpine and moist tundra were favorable for peatlands
initiation under a cooler climate. No forested peatlands formed before the HTM. Under the warm
condition in the HTM, boreal evergreen needleleaf and deciduous broadleaf forests expanded
(Figures 2b, c) as indicated by other studies (Bartlein et al., 2011; Edwards et al., 2005; Williams
et al., 2001). Meanwhile, large areas were taken up by forested peatlands, characterized by the



sharp increase of SOC storage in such ecosystems. The cooler temperature during the mid-
Holocene limited the productivity of tree plants, leading to the retreat of trees. This is broadly
consistent with other studies (Prentice et al., 1996; Edwards et al., 2000; Williams et al., 2001;
Bigelow et al., 2003). Large proportion of forested peatlands thus changed back into *Sphagnum*
spp. peatlands. The retreat of treeline on the Seward Peninsula in the cooler mid-Holocene likely
reflects much shorter and cooler growing seasons, influenced by an expansion of sea ice in the
Bering Sea (Crockford and Frederick, 2007) and the onset of the cooler Neoglacial climate.
Forested peatlands ceased accumulating SOC in central Alaska with an overall low accumulation
rates through the whole mid- to late-Holocene (Figures 8, 9, 11).
**4.3 Comparison between Simulated Carbon Dynamics and Other Estimates**
A large variation of "true" peat C accumulation rates was simulated on the Kenai
Peninsula (Figure 4a), ranging from -4 (that is, peat C loss) to 50 $C\ m^{-2}\ yr^{-1}$. We simulated an
average of peat SOC "apparent" accumulation rate of 11.4 g $C\ m^{-2}\ yr^{-1}$ from 15 to 5 ka (Figure
4b), which was slightly higher than the observed average rate from four sites (10.45
g $C\ m^{-2}\ yr^{-1}$). The simulated rate during the HTM was 26.5 g $C\ m^{-2}\ yr^{-1}$, up to five times
higher than the rest of the Holocene (5.04 g $C\ m^{-2}\ yr^{-1}$). The simulation results corresponded to
the observations, in which an average rate of 20 $C\ m^{-2}\ yr^{-1}$ from 11.5 to 8.6 ka was observed,
four times higher than 5 $C\ m^{-2}\ yr^{-1}$ over the rest of the Holocene.
We estimated an average peat SOC "apparent" accumulation rate of 13 g $C\ m^{-2}yr^{-1}$ (2.3
Tg $C\ yr^{-1}$ for the entire Alaska) from 15 ka to 2000 AD, lower than the value of 18.6
g $C\ m^{-2}yr^{-1}$ as estimated from peat cores for northern peatlands (Yu et al., 2010), and slightly
higher than the observed rate of 13.2 g $C\ m^{-2}yr^{-1}$ from four peatlands in Alaska (Jones and Yu,



2010). A simulated peak occurred during the HTM with the rate 29.1 g C $m^{-2}yr^{-1}$ (5.1 Tg C
$yr^{-1}$), which was slightly higher than the observed 25 g C $m^{-2}yr^{-1}$ for northern peatlands and
~20 g C $m^{-2}yr^{-1}$ for Alaska (Yu et al., 2010). It was almost four times higher than the rate 6.9
g C $m^{-2}yr^{-1}$ (1.4 Tg C $yr^{-1}$) over the rest of the Holocene, which corresponded to the peat core-
based observations of ~5 g C $m^{-2}yr^{-1}$. The mid- and late Holocene showed much slower C
accumulation at a rate approximately five folds lower than during the HTM. This corresponded
to the observation of a six-fold decrease in the rate of new peatland formation after 8.6 ka (Jones
and Yu 2010). The C accumulation rates increased abruptly to 39.2 g C $m^{-2}yr^{-1}$during the last
century, within the field-measured average apparent rate range of 20-50 g C $m^{-2}yr^{-1}$ over the
last 2000 years (Yu et al., 2010).

The SOC stock of northern peatlands has been estimated in many studies, ranging from

210 to 621 Pg (Oechel 1989; Gorham 1991; Armentano and Menges, 1986; Turunen et al., 2002;
Yu et al., 2010; see Yu 2012 for a review). Assuming Alaskan peatlands were representative of
northern peatlands and using the area of Alaskan peatlands ($0.45 \times 10^6$ $km^2$; Kivinen and
Pakarinen, 1981) divided by the total area of northern peatlnads (~$4 \times 10^6$ $km^2$; Maltby and
Immirzi 1993), we estimated a SOC stock of 23.6-69.9 Pg C for Alaskan peatlands. Our model
estimated 27-48 Pg C had been accumulated from 15 ka to 2000 AD. The uncertainty may be
resulted from peat basal age distributions and the peatland area, as we used modern inundation
data to estimate an area of $0.26 \times 10^6$ $km^2$. By incorporating the observed basal age distribution,
we estimated that approximately 68% of Alaskan peatlands had formed by the end of the HTM,
similar to the estimation from observed basal peat ages that 75% peatlands have formed by 8.6
ka (Jones and Yu 2010).



The northern circumpolar soils were estimated to cover approximately $18.78 \times 10^6$ km$^2$
(Tarnocai et al., 2009). The non-peatland soil C stock was estimated to be in the range of 150-
191 Pg C for boreal forests (Apps et al., 1993; Jobbagy and Jackson, 2000), and 60-144 Pg C for
tundra (Apps et al., 1993; Gilmanov and Oechel, 1995; Oechel et al., 1993) in the 0-100 cm
depth. Using the difference between Alaskan total land area ($1.69 \times 10^6$ km$^2$) and peatland area
($0.45 \times 10^6$ km$^2$), we estimated that the non-peatland area in Alaska was $1.24 \times 10^6$ km$^2$.
Therefore, Alaska non-peatland area contained 17-27 Pg C by using the ratio of Alaskan non-
peatland over northern non-peatland. In comparison, our estimate of 9-15 Pg C within 1-meter
depth suggested that our model might have underestimated the C stock for non-peatland soils.
Meanwhile, our estimated 2.5-3.7 Pg C stored in the Alaskan vegetation was lower than the
previous estimate of 5 Pg (Balshi et al., 2007; McGuire et al., 2009). The underestimation could
be resulted from the uncertainties in both peatland area fraction within each grid and the model
parameterization.
The simulated modern SOC distribution (Figure 12c) was largely consistent with the
study of Hugelius et al. (2014) (see Figure 3 in the paper). The model captured the high peat
SOC density areas on northern and southwestern coasts of Alaska, where observational data
showed some locations with SOC >75 kg C m$^{-2}$. This corresponded to our model simulation that
many grids had the SOC >75 kg C m$^{-2}$ in those areas. The observed overall average SOC
density of >40 kg C m$^{-2}$ was also consistent with our simulation. Eastern part and west coast had
the lowest SOC densities, corresponding to the model result that most grids in those areas had
SOC values between 20 and 40 kg C m$^{-2}$. Our estimated average peat depth of 1.9 m ranging
from 1.1 to 2.7 m from simulated peat SOC density was similar to the observed mean depth of
2.29 m for Alaskan peatlands (Gorham et al., 1991, 2012). Our estimates (Figure 12d) showed a




high correlation with the 64 observed peat samples (Figure 14) ($R^2 = 0.45$). The large intercept
of the regression line (101 cm) suggested that the model may not perform well in estimating the
grids with low peat depths (<50 cm).
**5. Conclusions**

We used a biogeochemistry model for both peatland and non-peatland ecosystems to

quantify the C stock and its changes over time in terrestrial ecosystems of Alaska during the last
15,000 years. The simulated peat SOC accumulation rates were compared with peat-core data
from four peatlands on the Kenai Peninsula in southern Alaska. The model well estimated the
peat SOC accumulation rates trajectory throughout the Holocene, indicating the model's
suitability for simulating peat C dynamics. Our regional simulation showed that 36-63 Pg C had
been accumulated in Alaskan land ecosystems since 15,000 years ago, including 27-48 Pg C in
peatlands and 9-15 Pg C in non-peatlands (within 1 m depth). We also estimated that 2.5-3.7 Pg
C was stored in contemporary Alaskan vegetation, with 0.3-0.6 Pg C in peatlands and 2.2-3.1 Pg
C in non-peatlands. The estimated average rate of peat C accumulation was 2.3 Tg C $\mathrm{yr}^{-1}$ with a
peak (5.1 Pg C $\mathrm{yr}^{-1}$) in the Holocene Thermal Maximum (HTM), four folds higher than the rate
of 1.4 Pg C $\mathrm{yr}^{-1}$ over the rest of the Holocene. The 20[th] century represented another high SOC
accumulation period after the much lowered accumulation rate in the late Holocene. We
estimated an average depth of 1.9 m of peat in Alaskan peatlands, similar to the observed mean
depth. We found that the changes of vegetation distribution due to the climatic change were the
key factors to the spatial variations of SOC accumulation in different time periods. The warming
in the HTM characterized by the increased summer temperature and increased seasonality of
solar radiation, along with the higher precipitation might have played an important role in



causing the high C accumulation rates. Under warming climate conditions, Alaskan peatlands
may continue acting as C sink in the future.
**6. Acknowledgment**. We acknowledge the funding support from a NSF project IIS-1027955
and a DOE project DE-SC0008092. We also acknowledge the SPRUCE project to allow us use
its data. Data presented in this paper are publicly accessible: ECBilt-CLIO Paleosimulation
(http://apdrc.soest.hawaii.edu/datadoc/sim2bl.php), CRU2.0 (http://www.cru.uea.ac.uk/data).
Model parameter data and model evaluation process are in Wang et al. (2016). Other simulation
data including model codes are available upon request from the corresponding author
(qzhuang@purdue.edu).

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



Table 1. Assignment of biomized fossil pollen data to the vegetation types in TEM (He et al.,
857  2014).

| TEM upland vegetation | TEM peatland vegetation | BIOMISE code |
|---|---|---|
| Alpine tundra | | CUSH   DRYT   PROS |
| Moist tundra | *Sphagnum* spp. open fen | DWAR  SHRU |
| Boreal evergreen needleleaf and mixed forest | *Sphagnum*-black spruce bog | TAIG   COCO   CLMX COMX |
| Boreal deciduous broadleaf forest | | CLDE |



Table 2. Relations between peatland basal age and vegetation distribution

| Peatland basal age | Vegetation types | Location |
|---|---|---|
| 15-11 ka | alpine tundra | south, northwestern, and southeastern coast |
| 11-10 ka | moist tundra | south, north, and southeastern coast |
| | boreal evergreen needleleaf forest | |
| | boreal deciduous broadleaf forest | east central part |
| 10-9 ka | moist tundra | south and north coast |
| | boreal evergreen needleleaf forest | central part |
| | boreal deciduous broadleaf forest | |
| 9-5 ka | moist tundra | central part |
| | boreal evergreen needleleaf forest | |
| 5 ka-1900 AD | moist tundra | west coast |
| | boreal evergreen needleleaf forest | |
















Table 3. Description of sites and variables used for parameterizing the core carbon and nitrogen
module (CNDM).

| Site[a] | Vegetation | Observed variables for CNDM parameterization | References |
|---|---|---|---|
| APEXCON | Moderate rich open fen with sedges (*Carex* sp.), spiked rushes (*Eleocharis* sp.), *Sphagnum* spp., and brown mosses (e.g., *Drepanocladus aduncus*) | Mean annual aboveground NPP in 2009; Mean annual belowground NPP in 2009; Aboveground biomass in 2009 | Chivers et al. (2009) Turetsky et al. (2008) Kane et al. (2010) Churchill et al. (2011) |
| APEXPER | Peat plateau bog with black spruce (*Picea mariana*), *Sphagnum* spp., and feather mosses | | |

[a]The Alaskan Peatland Experiment (APEX) site is adjacent to the Bonanza Creek Experimental Forest (BCEF) site,
approximately 35 km southwest of Fairbanks, AK. The area is classified as continental boreal climate with a mean annual
temperature of -2.9℃ and annual precipitation of 269 mm, of which 30% is snow (Hinzman et al., 2006).


Table 4. Carbon pools and fluxes used for calibration of CMDM

| Annual Carbon Fluxes or Pools[a] | *Sphagnum* Open Fen | | *Sphagnum*-Black Spruce Bog | | References |
|---|---|---|---|---|---|
| | Observation | Simulation | Observation | Simulation | Turetsky et al. (2008), Churchill (2011) Moore et al. (2002) Zhuang et al. (2002) Tarnocai et al. (2009) Kuhry and Vitt (1996) |
| NPP | 445±260 | 410 | 433±107 | 390 | |
| Aboveground Vegetation Carbon | 149-287 | | 423 | | |
| Belowground Vegetation Carbon | 564 | | 658-1128 | | |
| Total Vegetation Carbon Density | 713-851 | 800 | 732-1551 | 1300 | |
| Litter Fall Carbon Flux | 300 | 333 | 300 | 290 | |
| Methane Emission Flux | 19.5 | 19.2 | 9.7 | 12.8 | |


[a] Units for annual net primary production (NPP) and litter fall carbon are $g\ C\ m^{-2}\ yr^{-1}$. Units for vegetation carbon density are
$g\ C\ m^{-2}$. Units for Methane emissions are $g\ C - CH_4\ m^{-2}\ yr^{-1}$. The simulated total annual methane fluxes were compared with
the observations at APEXCON in 2005 and SPRUCE in 2012. A ratio of 0.47 was used to convert vegetation biomass to carbon
(Raich 1991).












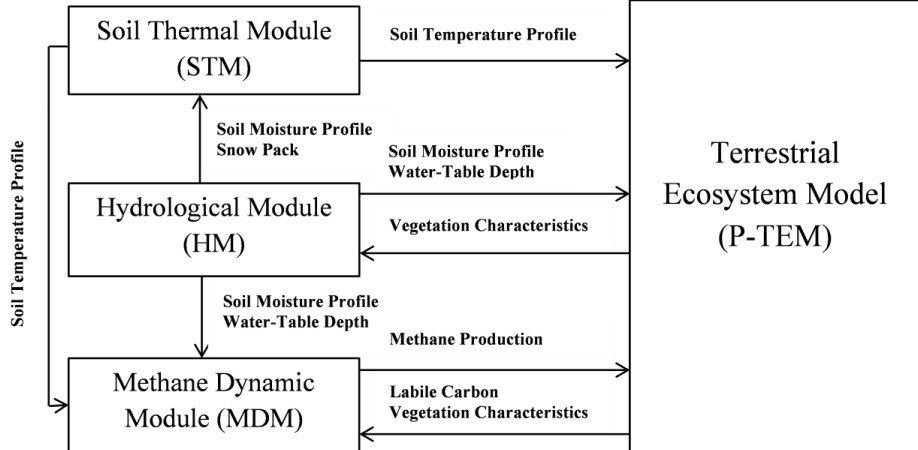

Figure 1. P-TEM (Peatland-Terrestrial Ecosystem Model) modeling framework, including a soil
thermal module (STM), a hydrologic module (HM), a carbon/ nitrogen dynamic model (CNDM),
and a methane dynamics module (MDM) (Wang et al., 2016).













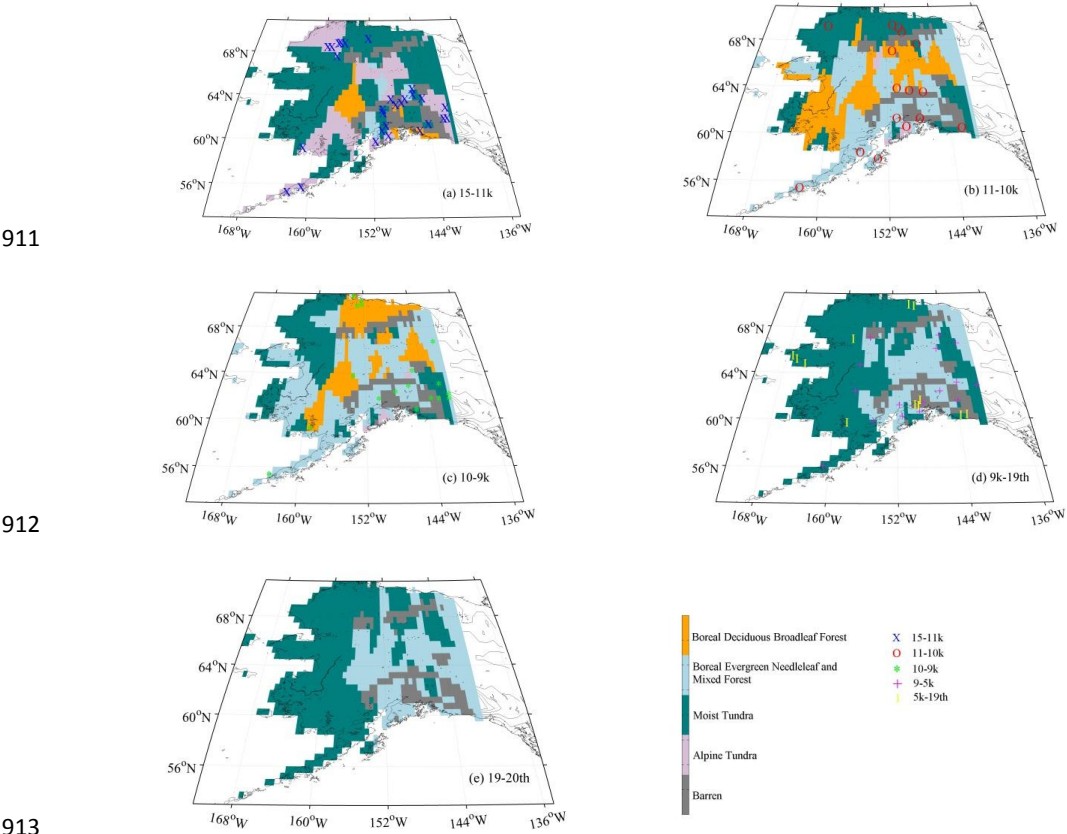




Figure 2. Alaskan vegetation distribution maps reconstructed from fossil pollen data during (a)
15-11 ka, (b) 11-10 ka, (c) 10-9 ka, (d) 9 ka -1900 AD, and (e) 1900-2000 AD (He et al., 2014).
Symbols represent the basal age of peat samples (n = 102) in Gorham et al. (2012). Barren
refers to mountain range and large body areas which could not be interpolated.






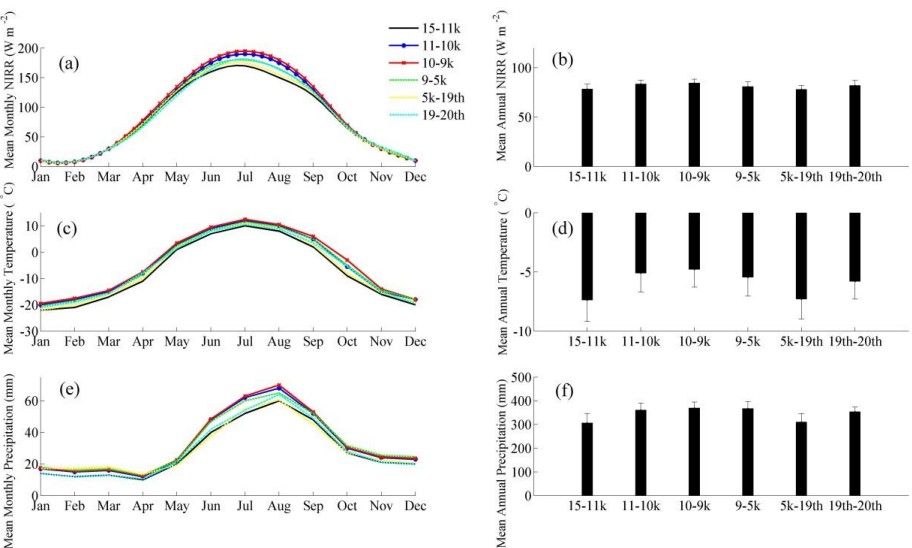

Figure 3. Simulated Paleo-climate and other input data from 15 ka to 2000 AD, including (a) mean monthly and (b) mean annual net incoming solar radiation (NIRR, W m$^{-2}$), (c) mean monthly and (d) mean annual air temperature (℃), (e) mean monthly and (f) mean annual precipitation (mm) (Timm and Timmermann, 2007; He et al., 2014).





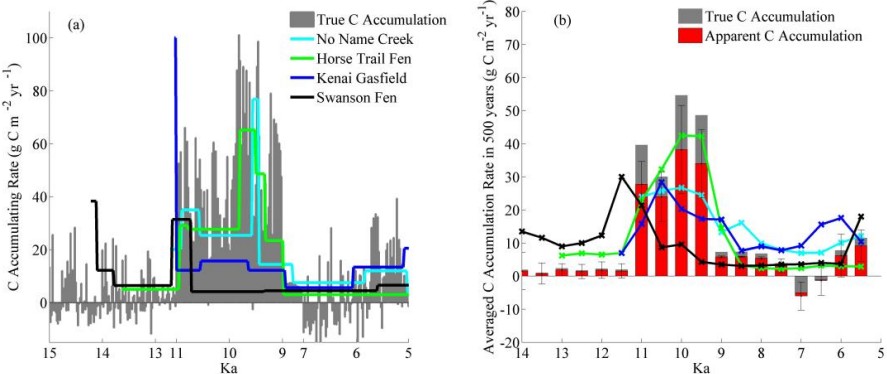

Figure 4. Simulated and observed carbon accumulation rates from 15 ka to 5 ka in 20-yr bins (a)
and 500-yr bins with standard deviation (b) for No Name Creek, Horse Trail Fen, Kenai Gasfield,
and Swanson Fen. Peat-core data were from Jones and Yu (2010).











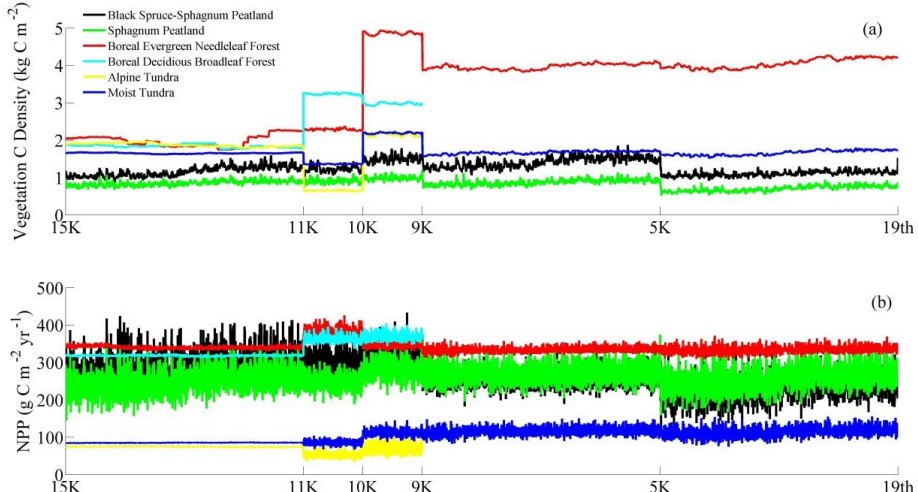

Figure 5. Simulated (a) mean vegetation carbon density (kg C m$^{-2}$) of different vegetation types
and (b) NPP (g C m$^{-2}$yr$^{-1}$).



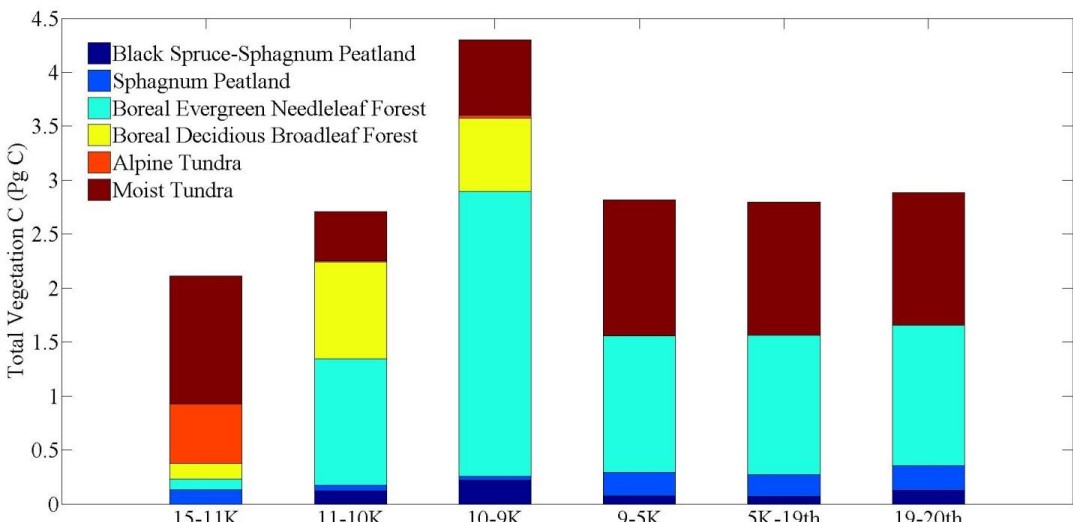

Figure 6. Total C (Pg C) stored in vegetation of Alaska for different time periods.





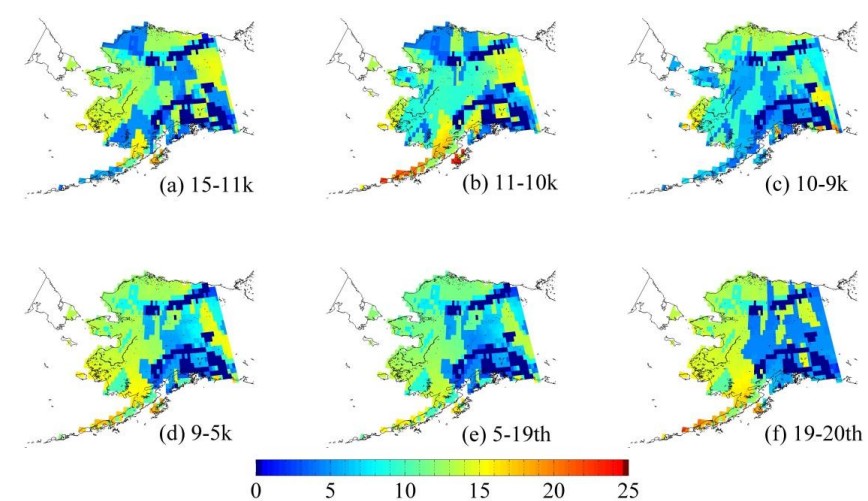


Figure 7. Non-peatland (mineral) SOC density (kg C m$^{-2}$) (cumulative) during (a) 15-11 ka, (b)
11-10 ka, (c) 10-9 ka, (d) 9-5 ka, (e) 5 ka -1900 AD, and (f) 1900-2000 AD.










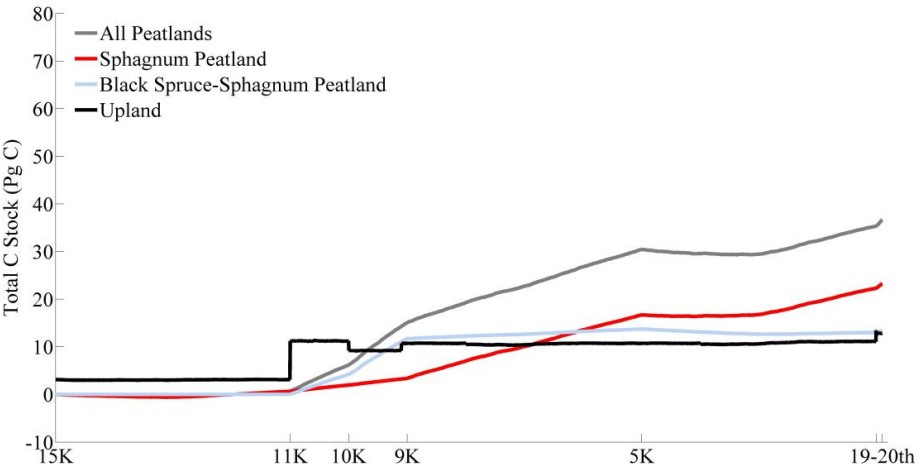


Figure 8. Total C stock accumulated from 15 ka to 2000 AD for all peatlands, *Sphagnum* open
peatland, *Sphagnum*-black spruce peatland, and upland soils.










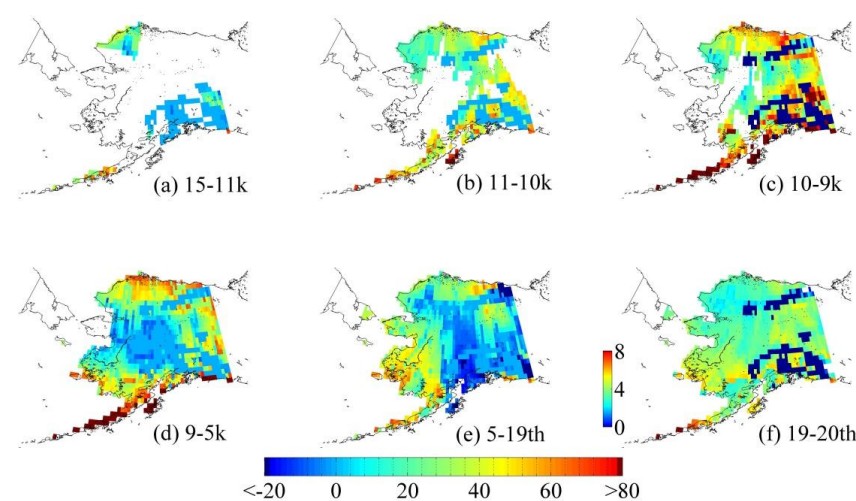

Figure 9. Peatland area expansion and peat soil C accumulation per 1000 years (kg C m$^{-2}$ kyr$^{-1}$)
during (a) 15-11 ka, (b) 11-10 ka, (c) 10-9 ka, (d) 9-5 ka, (e) 5 ka -1900 AD, and (f) 1900-2000
AD. The amount of C represents the C accumulation as the difference between the peat C
amount in the final year and the first year in each time slice.

















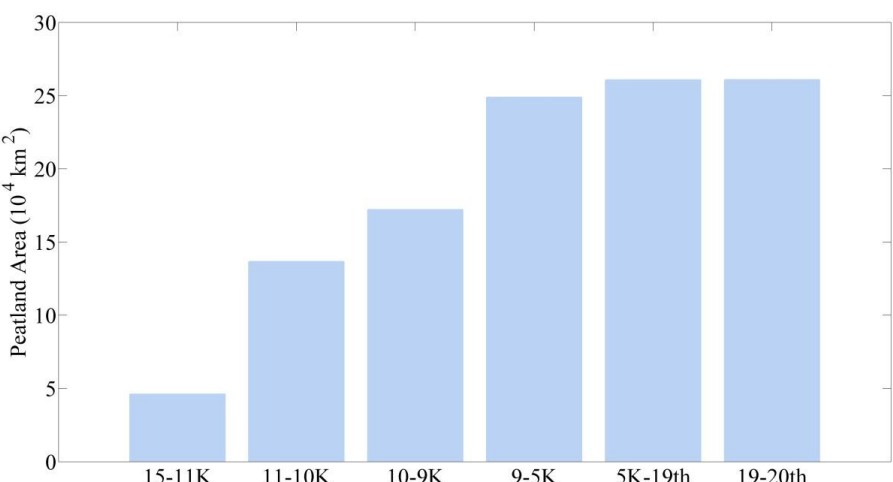


Figure 10. Peatland expansion area ($10^4$ km$^2$) in different time slices, the area of barren in the
map is set to 0 km$^2$.




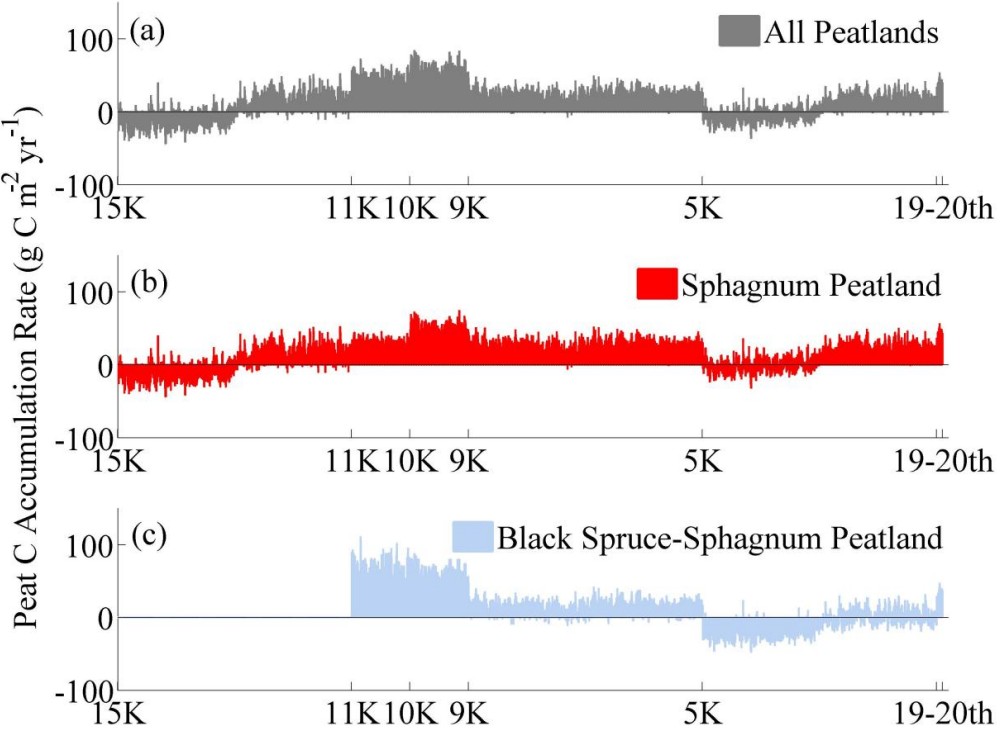


Figure 11. Peatland mean C accumulation rates from 15 ka to 2000 AD for (a) weighted average
of all peatlands, (b) *Sphagnum* open peatland, and (c) *Sphagnum*-black spruce peatland.













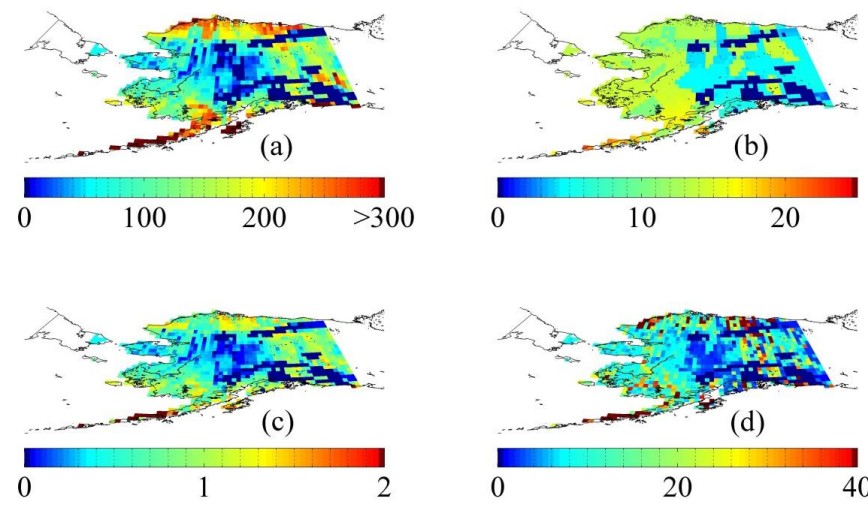

Figure 12. The spatial distribution of (a) total peat SOC density (kg C m$^{-2}$), (b) total mineral
SOC density (kg C m$^{-2}$), (c) total peat depth (m), and (d) weighted average of total (peatlands
plus non-peatlands) SOC density (kg C m$^{-2}$) in Alaska from 15 ka to 2000 AD.







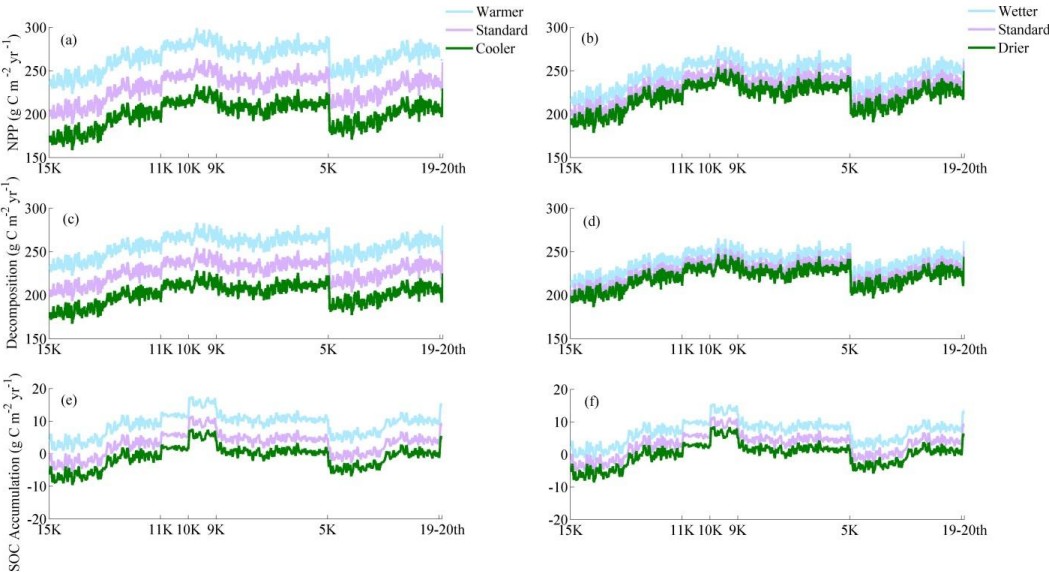


Figure 13. Temperature and precipitation effects on (a)(b) annual NPP, (c)(d) annual SOC
decomposition rate (aerobic plus anaerobic), and (e)(f) annual SOC accumulation rate of Alaska.
A 10-year moving average was applied.





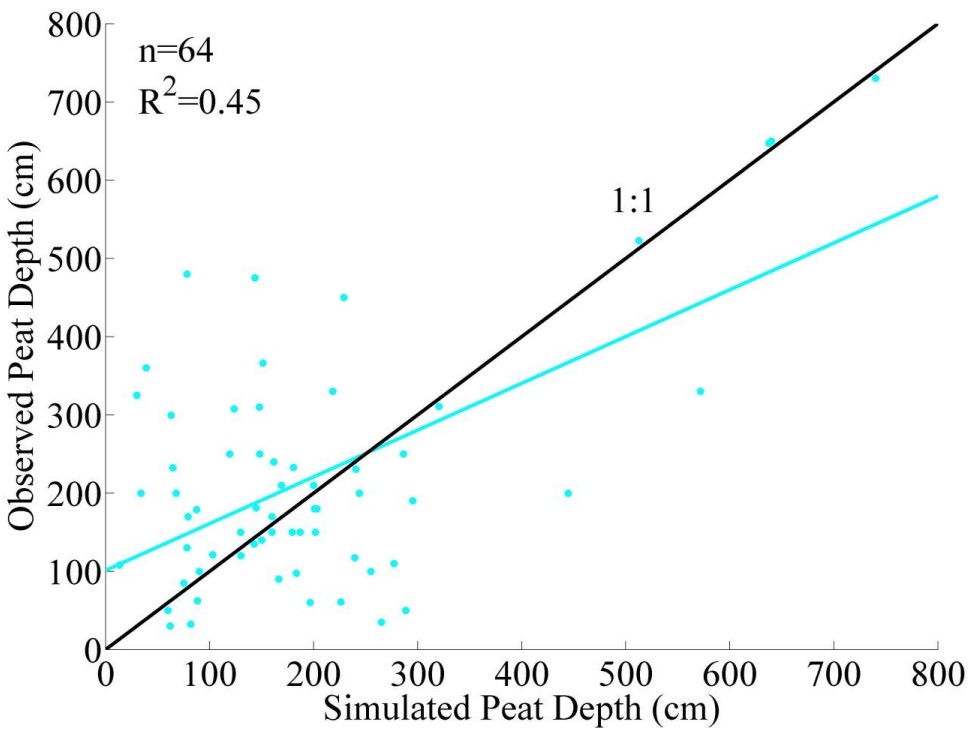

Figure 14. Field-based estimates and model simulations for peat depths in Alaska: The observed and simulated data are extracted from the same grids on the map. Linear regression line (cyan) is compared with the 1:1 line. The linear regression is significant (P<0.001, n = 64) with $R^2 = 0.45$, slope = 0.65, and intercept = 101.05 cm. The observations of >1000 cm are treated as outliers.