# Peer review of "Quantifying Soil Carbon Accumulation in Alaskan Terrestrial Ecosystems during the Last 15,000 Years Sirui Wang1, Qianlai Zhuang1,2\*, Zicheng Yu3 1Department of Earth, Atmospheric, and Planetary Sciences, Purdue University, West Lafayette, Indiana, 47907 <"

_Biogeosciences, 2016_

## Referee Comment (RC1) · T. Kleinen (Referee) · 15 Aug 2016

In their manuscript, the authors present a model study of soil carbon accumulation in Alaska over the last 15000 years with a special focus on peat carbon accumulation.

Overall, the manuscript is quite interesting and suitable for publication in biogeosciences. However, a number of minor issues remain before it is ready for publication.

1. Figure 2 shows the vegetation distribution used to drive the model for 5 time slices. However, results are presented in Figs. 7 and 9 for 6 time slices. This is confusing for the reader. At the very least it needs to be clearly marked in the figure caption. The symbols used in Fig. 2 to show peat basal dates are also very difficult to make out. Maybe it is possible for the authors to make this Figure clearer.

[Figure]

2. Figure 7: Figure caption is unclear. How can you show cumulative SOC density? This would imply only the very last time in the time interval is shown. I assume you actually mean the mean SOC density.

3. Figure 12: Caption also unclear. I assume the late 20th century distribution is shown? Figure caption shows 15ka to 2000AD, implying a mean value over this time frame.

4. Since a large part of the results hinges on the modelled changes in peatland area, it is essential that a description of how area changes are determined is provided. Currently it is only stated that area is prescribed from Matthews & Fung, implying no change is possible.

5. Page 4, lines 94-95: the Spahni et al. Model has actually been evaluated with respect to the variables listed – see Wania et al. Publications on the LPJ-Why model on which Spahni is based.

6. Page 5, line 97: Why do you cite Kleinen et al. 2010? They do not use a process-based peatland model, but rather prescribed peat accumulation. I assume you actually meant to cite Kleinen et al. 2012?

7. Page 7 and 8, lines 159-166: The aboveground vegetation in your calibration site is significantly different from the Mer Bleu site you use for belowground calibration. In addition the climatic situation at the two sites is significantly different. Therefore is seems quite a stretch to argue that belowground processes are basically the same. Please provide more justification for this assumption.

8. Page 8, line 173: Please correct date for late Holocene time frame

9. Page 9, lines 194-195: The Shuttle Radar Topography Mission (SRTM) only covered latitudes 56S-60N. Therefore there is NO SRTM data for Alaska. You obviously used some other source for topography data – please provide correct reference.

10. Page 9, lines 197-203: Downscaling / bias correction is unclear. From the text one

gets the impression, that ECBILT fields and CRU data may be significantly different for 20th century. However, my reading of the original publications is that bias correction minimised that difference. Please clarify this – it would strongly strengthen the text.

11. Page 12, line 256: References to Figures 2 and 3 mixed up, please correct.

12. Page 13, lines 279-289: Study sites unclear. Please provide table of site locations.

13. Page 14, lines 314-316 and table 4: table 4 only lists uncertainty ranges for peatland vegetation. How were uncertainty ranges for upland vegetation derived? Certainly not from the ranges in table 4. Please clarify.

14. Page 15, lines 333-334 – see comment #13.

15. Page 15, line 339 – page 16, line 349: how were peatland area changes determined? Completely unclear (see comment #4)

16. Thankfully, the authors have used a spellchecker, so there are very few typos. However, there are numerous places in the text, where grammar needs checking: Temporal forms are not always consistent, and some sentences are missing single words or larger parts. Therefore CAREFUL COPY-EDITING is highly important.

---

## Referee Comment (RC2) · Dr Tupek (Referee) · 25 Aug 2016

Review of Wang et al. "Quantifying Soil Carbon Accumulation in Alaskan Terrestrial Ecosystems during the Last 15,000 Years"

=======

Summary

=======

Authors tested SOC simulations from the modified version of their Terrestrial Ecosystem Model (TEM) for peatlands (P-TEM) for Alaska region for time periods defined by climatic characteristics (solar radiation, temperature, and precipitation levels) and vegetation distribution during last 15 000 years. The model was applied for peatlands and

non-peatlands (mineral soil forests). Simulated C distributions, NPP, and peat depths were presented for Alaska at fine scale spatial resolution (maps) and summarized for vegetation types. This can be an interesting study if presented carefully.

============

General comments

============

The advances of the model include hydrology, soil thermal, C and N dynamics modules. This looks promising however, the model code is not publicly available and need to be asked from authors. No software details were given. I'd like to see the code and run an example simulation in R to understand the model structure.

Description of NPP simulation by the model is missing. Description of Carbon and Nitrogen dynamics of CNDM module is also missing. Add descriptions for clarity. Results largely depended on the adopted distribution of vegetation types. Authors mentioned that distribution of vegetation types and their changes overdriven C accumulation over climate, but also noted that climate had probably driven distribution of vegetation types. Individual parameter values are not listed. It would be interesting to see the changes of key parameters between vegetation types in relation to their prevailing climate.

Description of observations of peat depths that were used for model validation is not sufficient. Some description is in section 2.5 but it is not clear and the points on the maps in Fig. 2 are barely visible. Describe clearly.

Structure of the paper is unclear. Reorder the ideas, avoid using repetitions.

Methods are presented in results. Results are presented in discussion. For example lines 369-370 in results describe how peat depth was calculated for the first time. Discussion Section 4.3 presents too many numbers without deeper insights on reason behind differences between other studies. At the end of discussion a scatterplot Fig. 14 between observed and modeled peat depths is presented for the 1st time. The Fig.

14 shows that without exceptional agreement of 3 largest values the rest of the scatter is just a gunshot indicating poor performance of the model in most conditions. Authors avoid the explanation. Move results to result section. Present some values in Tables. In discussion interpret the results with a focus on the model and data input.

Interesting results as underestimation in uplands, lack of C loss simulation (Fig. 2), reasons behind vegetation controlling C storage, disagreement with observations, assumption that peatlands will remain C sink are brushed away. The agreement with other studies is OK but not enough for discussion. Describe reasons for agreement/disagreements, give insights on function/performance and reason why to use/trust your model. Although the authors claimed that the PTEM includes CN module, nothing can be learned from reading the paper how this or other modules affect the results. Given the SOC underestimation of uplands and large scatter with peatlands, and large-scale climate estimates, could accounting for differences in nutrient status or reevaluating response of C/N ratio be a key for improved estimation of spatial variability of SOC accumulation of P-TEM or TEM model?

Authors claim that recent climate is warmer and wetter in summers and therefore with future warmer-wetter climate peatland carbon sink will continue. Possibility of increased respiration and C loss due to droughts or warmer winters is not mentioned. For the conclusion on future C sink a simulations with climate scenarios would be useful.

The interesting results are damped by poor presentation. The manuscript read more as an early draft, rather than discussion paper ready for peer-review. Awkward English throughout makes it difficult to follow the main message and to review scientific merits. Manuscript was probably rejected elsewhere, as the references are not corresponding to the format required by Biogeosciences. In spite of interesting modeling approach, without careful language editing and major content corrections the manuscript cannot stand as a scientific paper in Biogeosciences and should be rejected. That would be a shame because otherwise it can be a useful paper for developing accurate models for

peatland SOC accumulation.

============

Specific comments:

============

Increase Figs. 2, 3, 4 and their legends size by 50 or 100%

why are panels b, d, f in Fig. 3 scaled by zero? that makes differences to appear smaller increase legend in Fig. 5

increase Fig. 7 and 9, why fig. 9 it has 2 legend bars?

line 137 "observed water contents drive STM"?, did you mean observed temperatures?

lines 235-238 reformulate for clarity

line 293 correct value of C storage

lines 300-302, 309 reformulate for clarity

line 315 range of what?

lines 325 "spots were widely spread" reformulate, "SOC concentration" do you mean SOC storage?

line 326 reformulate "tundra was taking back area" or similar

line 361 Table 4 is not showing parameters

lines 375-377 reformulate for clarity

lines 420 – 460 reformulate for clarity

line 424 why if $p < 0.05$ "some certain effects"?

line 428 "positive effect" of temperature? low temp slowed SOC accumulation, that's

negative effect

line 437 delete "suggesting the warmest climate during HTM" it comes by definition of HTM

lnes 443-448 does not make sense

line 452 what is "stored C in overall in the spatial scale"?

line 454 "negative accumulation rate" avoid writing nonsense

lines 485 – 545 reformulate whole section, move results into results, check for repetitions, highlight only most important trends and insights, shorten discussion if nothing much relevant to say

lines 508-519 OK

References – change into the corresponding format of Biogeosciences

---

## Author Comment (AC1) · 14 Sep 2016

**Response to Referee #1 (Dr. Kleinen)**

We would like to thank Dr. Kleinen for his thoughtful and constructive review. Our responses to all of the referee's comments are italicized below.

*In their manuscript, the authors present a model study of soil carbon accumulation in Alaska over the last 15000 years with a special focus on peat carbon accumulation. Overall, the manuscript is quite interesting and suitable for publication in biogeosciences. However, a number of minor issues remain before it is ready for publication.*

We agree to make all edits and changes brought up by the referee. We respond to the comments below.

*Figure 2 shows the vegetation distribution used to drive the model for 5 time slices. However, results are presented in Figs. 7 and 9 for 6 time slices. This is confusing for the reader. At the very least it needs to be clearly marked in the figure caption. The symbols used in Fig. 2 to show peat basal dates are also very difficult to make out. Maybe it is possible for the authors to make this Figure clearer.*

We have five vegetation distribution maps for five time slices. In Fig. 2, we present those five maps. We present six maps of carbon density distribution in Figs. 7 and 9. The reason is that during 5 ka-1900 AD, there was climate change affecting carbon accumulation rates, although the vegetation map remains the same during 9ka-1900 AD. Yes, it is indeed confusing for readers. We agree to make explanation in the figure caption to clarify this. Also, we agree to make all the figures clearer and change the symbol size much larger.

*Figure 7: Figure caption is unclear. How can you show cumulative SOC density? This would imply only the very last time in the time interval is shown. I assume you actually mean the mean SOC density.*

Correct. For non-peatland soils, we want to show the mean carbon density during each time slices, as there is no significant change from period to period. We agree to make the caption clearer to readers and change the "cumulative" to "mean".

*Figure 12: Caption also unclear. I assume the late 20th century distribution is shown? Figure caption shows 15ka to 2000AD, implying a mean value over this time frame.*

The maps of total SOC density in Alaska are the sums of SOC during all periods, from 15ka, which is the beginning of the simulation, to late 20[th], which is the end. This is the cumulative carbon density, which is the amount at the very last year. We will make the caption much clearer.

*Since a large part of the results hinges on the modelled changes in peatland area, it is essential that a description of how area changes are determined is provided. Currently it is only stated that area is prescribed from Matthews & Fung, implying no change is possible.*

*Page 15, line 339 – page 16, line 349: how were peatland area changes determined? Completely unclear (see comment #4)*

The change of peatland area is determined by the basal ages of the peatland. And from the distribution of the basal ages in Fig. 2, we link some vegetation types during each time slice to the ages. Say, during 15-11ka, the peatland was formed based on the alpine tundra (Table 2). However, within each pixel, we assume the inundated area represents the peatland area and other area represents non-peatland. This is from the modern inundation map and does not change. The amount of pixels which have peatland vegetation types is changing through time determined by the basal age distribution. The percentage of peatland cover within each pixel is unchanging. We agree to discuss more on this.

*Page 4, lines 94-95: the Spahni et al. Model has actually been evaluated with respect to the variables listed – see Wania et al. Publications on the LPJ-Why model on which Spahni is based.*
Thanks for pointing out the wrong statement here. We agree to check the related references and make changes of the statement.

*Page 5, line 97: Why do you cite Kleinen et al. 2010? They do not use a processbased peatland model, but rather prescribed peat accumulation. I assume you actually meant to cite Kleinen et al. 2012?*

We will cite Kleinen et al. (2012) instead.

*Page 7 and 8, lines 159-166: The aboveground vegetation in your calibration site is significantly different from the Mer Bleu site you use for belowground calibration. In addition the climatic situation at the two sites is significantly different. Therefore is seems quite a stretch to argue that belowground processes are basically the same. Please provide more justification for this assumption.*

Assuming the belowground carbon in Mer Bleu is the same as that in APEXCON would be wrong. We agree to make further justification and if necessary, we will replace this value by a more persuasive one and re-calibrate the model.

*Page 8, line 173: Please correct date for late Holocene time frame*

We will correct the date from "9 ka-1900AD" to "5ka-1900 AD".

*Page 9, lines 194-195: The Shuttle Radar Topography Mission (SRTM) only covered latitudes 56S-60N. Therefore there is NO SRTM data for Alaska. You obviously used some other source for topography data – please provide correct reference.*

We directly used the data from He et al. (2014). We agree to find the correct source of the elevation data and will cite the correct reference.

*Page 9, lines 197-203: Downscaling / bias correction is unclear. From the text one gets the impression, that ECBILT fields and CRU data may be significantly different for 20th century. However, my reading of the original publications is that bias correction minimised that difference. Please clarify this – it would strongly strengthen the text.*

Yes, the climate data were bias-corrected to minimize the difference between ECBILT and CRU. We agree to make clarification on this.

*Page 12, line 256: References to Figures 2 and 3 mixed up, please correct.*

We will change the sequences of the references to correct such mistake.

*Page 13, lines 279-289: Study sites unclear. Please provide table of site locations.*

Below is the table of descriptions of those sites. We agree with adding this table or citing it in Wang et al. (2016) JGR-Biogeosciences paper.

Sites used for comparison of carbon accumulation rates between simulation and observation [*Jones and Yu,* 2010]

| Site name | Location | Peatland type | Latitude | Longitude | Dating method | No. of dates | Basal age (cal yr BP) | Time-weighted Holocene accumulation rates (g C m$^{-2}$ yr$^{-1}$) |
|---|---|---|---|---|---|---|---|---|
| Kenai Gasfield | Alaska, USA | fen | 60°27'N | 151°14'W | AMS | 12 | 11,408 | 13.1 |
| No Name Creek | Alaska, USA | fen | 60°38'N | 151°04'W | AMS | 11 | 11,526 | 12.3 |
| Horsetrail fen | Alaska, USA | rich fen | 60°25'N | 150°54'W | AMS | 10 | 13,614 | 10. 7 |
| Swanson fen | Alaska, USA | poor fen | 60°47'N | 150°49'W | AMS | 9 | 14,225 | 5.7 |

*Page 14, lines 314-316 and table 4: table 4 only lists uncertainty ranges for peatland vegetation. How were uncertainty ranges for upland vegetation derived? Certainly not from the ranges in table 4. Please clarify. Page 15, lines 333-334 – see comment #13.*

*Thankfully, the authors have used a spellchecker, so there are very few typos. However, there are numerous places in the text, where grammar needs checking: Temporal forms are not always consistent, and some sentences are missing single words or larger parts. Therefore CAREFUL COPY-EDITING is highly important.*

The uncertainties of the upland vegetation are from the uncertainties of parameters in previous study (Tang and Zhuang, 2008; 2009). We will clarify this and add the ranges of parameters.

---

## Author Comment (AC2) · 14 Sep 2016

**Response to Referee #2 (Dr. Tupek)**

We would like to thank Dr. Tupek for his thoughtful and constructive review, as well as his detailed comments. Our responses to all of the referee's comments are italicized below.

*Authors tested SOC simulations from the modified version of their Terrestrial Ecosystem Model (TEM) for peatlands (P-TEM) for Alaska region for time periods defined by climatic characteristics (solar radiation, temperature, and precipitation levels) and vegetation distribution during last 15 000 years. The model was applied for peatlands and non-peatlands (mineral soil forests). Simulated C distributions, NPP, and peat depths were presented for Alaska at fine scale spatial resolution (maps) and summarized for vegetation types. This can be an interesting study if presented carefully*

We agree to make all edits and changes brought up by the referee. We respond to the comments below.

*The advances of the model include hydrology, soil thermal, C and N dynamics modules. This looks promising however, the model code is not publicly available and need to be asked from authors. No software details were given. I'd like to see the code and run an example simulation in R to understand the model structure. Description of NPP simulation by the model is missing. Description of Carbon and Nitrogen dynamics of CNDM module is also missing. Add descriptions for clarity. Results largely depended on the adopted distribution of vegetation types. Authors mentioned that distribution of vegetation types and their changes overdriven C accumulation over climate, but also noted that climate had probably driven distribution of vegetation types. Individual parameter values are not listed. It would be interesting to see the changes of key parameters between vegetation types in relation to their prevailing climate.*

The descriptions of P-TEM including modules of STM, HM, CNDM, and MDM are less listed in this manuscript. The main reason is that we have described the model framework, specific parameters and the parameterization processes, along with the comparison between simulation and observation regarding soil moisture, methane emissions, water-table depth, and carbon and nitrogen dynamics in the previous study. Please find the paper "**Quantifying Peat Carbon Accumulation in Alaska Using a Process-Based Biogeochemistry Model**" for details (*Wang, S., Q. Zhuang, Z. Yu, S. Bridgham, and J. K. Keller (2016), Quantifying peat carbon accumulation in Alaska using a process-based biogeochemistry model, J. Geophys. Res. Biogeosci., 121, doi:10.1002/2016JG003452.*).

The paper was under review by JGR-Biogeosciences at that moment so there was no way we could cite that paper. However, it was just published and we have cited it during the revision of this manuscript. Besides, as an introduction to a newly built model (P-TEM), that JGR paper cited other references which have all the individual modules and their applications discussed in detail (e.g., see Zhuang et al., 2001 JGR, Jiang, Y. et al., 2012 JGR for STM module; Zhuang et al., 2002 JGR for HM; Zhuang et al., 2004 GBC for MDM; JW Raich et al., 1991; 1992 for CNDM). Also, please find all references with Terrestrial Ecosystem Model in use at: http://www.eaps.purdue.edu/ebdl/publications/index.html

*Description of observations of peat depths that were used for model validation is not sufficient. Some description is in section 2.5 but it is not clear and the points on the maps in Fig. 2 are barely visible. Describe clearly.*

The manuscript highlights the method and the result, however, has less emphasis on the model validation with observed peat depth distribution. We strongly agree to add more contents discussing the model validation and the use of observed peatland basal ages, as mentioned in section 2.5. We also agree to change the symbol size in Fig. 2 to make the figure much clearer to readers.

*Structure of the paper is unclear. Reorder the ideas, avoid using repetitions.*

We will reorder the ideas and make it more logical to readers.

*Methods are presented in results. Results are presented in discussion. For example lines 369-370 in results describe how peat depth was calculated for the first time. Discussion Section 4.3 presents too many numbers without deeper insights on reason behind differences between other studies. At the end of discussion a scatterplot Fig. 14 between observed and modeled peat depths is presented for the 1st time. The Fig. 14 shows that without exceptional agreement of 3 largest values the rest of the scatter is just a gunshot indicating poor performance of the model in most conditions. Authors avoid the explanation. Move results to result section. Present some values in Tables. In discussion interpret the results with a focus on the model and data input.*

*Interesting results as underestimation in uplands, lack of C loss simulation (Fig. 2), reasons behind vegetation controlling C storage, disagreement with observations, assumption that peatlands will remain C sink are brushed away. The agreement with other studies is OK but not enough for discussion. Describe reasons for agreement/disagreements, give insights on function/performance and reason why to use/trust your model. Although the authors claimed that the PTEM includes CN module, nothing can be learned from reading the paper how this or other modules affect the results. Given the SOC underestimation of uplands and large scatter with peatlands, and large-scale climate estimates, could accounting for differences in nutrient status or reevaluating response of C/N ratio be a key for improved estimation of spatial variability of SOC accumulation of P-TEM or TEM model?*

Many results and conclusions were obtained from this study. We agree to reorder the structure and make the results more concise and focused. Tables will be added to present the findings more clearly. The discussion section will be reorganized and results will be analyzed in detail. We will put more discussion on model validation and the relationship of vegetation distribution, climate change, and carbon accumulation rates. We will also cite the newly published JGR paper as mentioned above to give readers a clearer vision on P-TEM model framework and its application to Alaskan peatland.

*Authors claim that recent climate is warmer and wetter in summers and therefore with future warmer-wetter climate peatland carbon sink will continue. Possibility of increased respiration and C loss due to droughts or warmer winters is not mentioned. For the conclusion on future C sink a simulations with climate scenarios would be useful.*

We will consider the simulation using RCP data. The simulation requires data preparation and organization. This may take a relatively long time to complete. The discussion section may be lengthened to include future scenarios.

*Increase Figs. 2, 3, 4 and their legends size by 50 or 100%*

We agree to increase the legends size and symbol size to make the figures clear to see.

*why are panels b, d, f in Fig. 3 scaled by zero? that makes differences to appear smaller increase legend in Fig. 5*

Agreed. We should decrease the axis scale in Fig. 3 to make the difference appear bigger. We will increase the legend in Fig. 5.

*increase Fig. 7 and 9, why fig. 9 it has 2 legend bars?*

We agree to increase the size of each panel in Fig. 7 and Fig. 9. Fig. 9 represents the carbon accumulation amount during each time slice. The unit is kg C m$^{-2}$ kyr$^{-1}$. We mistakenly put extra color bar in panel (f). We will delete it.

*line 137 "observed water contents drive STM"?, did you mean observed temperatures?*

The STM considers soil water content at different depth to simulate the soil temperature. We used the observed soil water content at particular site to drive the STM when parameterizing the model in *Wang, S., Q. Zhuang, Z. Yu, S. Bridgham, and J. K. Keller (2016), Quantifying peat carbon accumulation in Alaska using a process-based biogeochemistry model, J. Geophys. Res. Biogeosci., 121, doi:10.1002/2016JG003452.*. During the regional simulation, we used the simulated soil water content at different depth directly from HM to drive the STM.

*lines 235-238 reformulate for clarity*

We compared the peatland basal age distribution with the vegetation types during each time slice and found some relationships among them. We will make specific statements on this.

*line 293 correct value of C storage*

We will change this value to 0.8 kg C m$^{-2}$.

*lines 300-302, 309 reformulate for clarity*

We will make the statement clearer and more concise.

*line 315 range of what?*

The range here is for vegetation carbon storage. We will clarify this.

*lines 325 "spots were widely spread" reformulate, "SOC concentration" do you mean SOC storage?*

We will change it into "there were fewer spots with low SOC density". The "SOC concentration" is SOC storage or density for particular pixel.

*line 326 reformulate "tundra was taking back area" or similar*

We will change it to "tundra area increased" or similar.

*line 361 Table 4 is not showing parameters*

Agreed. We will change the statement to "due to uncertainties of observation" or similar.

*lines 375-377 reformulate for clarity*

We will reformulate the sentences as "The pixels with highest SOC density were mainly located in northern coast. Southwestern coast and eastern central also had high carbon storage (>40 kg C m$^{-2}$), while areas with lowest density (<20 kg C m$^{-2}$) were in central and western parts. *lines 420 – 460 reformulate for clarity*

We will reformulate the discussion part.

*line 424 why if p < 0.05 "some certain effects"?*

From the result in Wang et al. (2016) that p-value is less than 0.05, we can get the idea that the interaction factor does have some effects on the carbon accumulation rates. We will clarify this.

*line 428 "positive effect" of temperature? low temp slowed SOC accumulation, that's negative effect*

Yes, it should be "negative effect" as the cooler condition during the neoglacier period slowed the carbon accumulation. We will clarify it.

*line 437 delete "suggesting the warmest climate during HTM" it comes by definition of HTM*

Correct. Repeating the definition of HTM can be awkward. We will revise it.

*lines 443-448 does not make sense*

We will delete this part since it is talking about the uncertainties coming from the climate model (ECBILT) itself, which can be tedious and less focused. We directly used the output of the model.

*line 452 what is "stored C in overall in the spatial scale"?*

We should have made the sentence clear. We will change it into "Overall, SBP stored lower C than SP".

*line 454 "negative accumulation rate" avoid writing nonsense*

We will change it to "with some areas accompanied by C loss".

*lines 485 – 545 reformulate whole section, move results into results, check for repetitions, highlight only most important trends and insights, shorten discussion if nothing much relevant to say*

We will follow your and editor's comments to make the revision.

*lines 508-519 OK*

*References – change into the corresponding format of Biogeosciences*

We will change all the references into the format of Biogeosciences.

We planned to submit the manuscript to JGR-Biogeoscience at first. In this manuscript, we cited Wang et al. (2016) JGR paper, which was under revision at that moment. We thought it might complicate the editor's decision. Therefore we decided to submit it to Biogeosciences. However, the manuscript has never been submitted to other journals before.

---

## Author Response (AR1)

**Response to Referee #1 (Dr. Kleinen)**

We would like to thank Dr. Kleinen for his thoughtful and constructive review. Our responses to all of the referee's comments (italicized) are presented below.

*In their manuscript, the authors present a model study of soil carbon accumulation in Alaska over the last 15000 years with a special focus on peat carbon accumulation. Overall, the manuscript is quite interesting and suitable for publication in biogeosciences. However, a number of minor issues remain before it is ready for publication.*

We made all edits and changes following the referee's comments and suggestions. We detailed our responses below.

*Figure 2 shows the vegetation distribution used to drive the model for 5 time slices. However, results are presented in Figs. 7 and 9 for 6 time slices. This is confusing for the reader. At the very least it needs to be clearly marked in the figure caption. The symbols used in Fig. 2 to show peat basal dates are also very difficult to make out. Maybe it is possible for the authors to make this Figure clearer.*

We have five vegetation distribution maps for five time slices. In Fig. 2, we present those five maps. We present six maps of carbon density distribution in Figs. 7 and 8. The reason is that during 5 ka-1900 AD, there was climate change affecting carbon accumulation rates, although the vegetation map remains the same during 9ka-1900 AD. Yes, it is indeed confusing for readers. Thus, in this revision, we made the explanation in the captions of Figs 2, 7, and 8 to clarify this. Also, we have made all panels in Fig. 2 clearer and changed the symbol size.

*Figure 7: Figure caption is unclear. How can you show cumulative SOC density? This would imply only the very last time in the time interval is shown. I assume you actually mean the mean SOC density.*

Correct. For non-peatland soils, we wanted to show the mean carbon density during each time slice. We have made the caption clearer to readers and change the "cumulative" to "average".

*Figure 12: Caption also unclear. I assume the late 20th century distribution is shown? Figure caption shows 15ka to 2000AD, implying a mean value over this time frame.*

The maps of total SOC density in Alaska are the sums of SOC during all periods, from 15ka, which is the beginning of the simulation, to the late 20$^{\text{th}}$, which is the end. This is the cumulative carbon density. We have changed the caption of panel (d) to "area-weighted total (peatland plus non-peatland) SOC density from 15 ka to 2000 AD, and added "total" in the descriptions of other panels.

*Since a large part of the results hinges on the modelled changes in peatland area, it is essential*
*that a description of how area changes are determined is provided. Currently it is only stated*
*that area is prescribed from Matthews & Fung, implying no change is possible.*

*Page 15, line 339 – page 16, line 349: how were peatland area changes determined? Completely*
*unclear (see comment #4)*

The change of peatland area is determined by the basal ages of the peatland. And from the
distribution of the basal ages in Fig. 2, we link some vegetation types during each time slice to
the ages. For example, during 15-11ka, the peatland area is determined based on the alpine
tundra. However, within each pixel, we assume the inundated area represents the peatland area
and other area represents non-peatland. The inundation area information is from the modern
inundation map and does not change. The number of pixels that have peatland vegetation is
changing through time determined by the basal age distribution. The percentage of peatland area
within each pixel is unchanging. We have made the description clearer in the Method section.
We added the peatland expansion into Results& Discussion section to better describe the link
between basal ages and the peatland area change. We also added the drawbacks of such
assumption.

*Page 4, lines 94-95: the Spahni et al. Model has actually been evaluated with respect to the*
*variables listed – see Wania et al. Publications on the LPJ-Why model on which Spahni is based.*

Thanks for pointing out our wrong statement here. We checked the relevant references and
corrected our statement.

*Page 5, line 97: Why do you cite Kleinen et al. 2010? They do not use a processbased peatland*
*model, but rather prescribed peat accumulation. I assume you actually meant to cite Kleinen et*
*al. 2012?*

We have cited Kleinen et al. (2012) instead.

*Page 7 and 8, lines 159-166: The aboveground vegetation in your calibration site is significantly*
*different from the Mer Bleu site you use for belowground calibration. In addition the climatic*
*situation at the two sites is significantly different. Therefore is seems quite a stretch to argue that*
*belowground processes are basically the same. Please provide more justification for this*
*assumption.*

Assuming the belowground carbon in Mer Bleu is the same as that in APEXCON could be
wrong. In this revision, we added another site (Suurisuo mire complex) in southern Finland,
which is a sedge fen dominated by *Carex rostrate.* We compared the ratios of belowground
biomass over total biomass at both sites and applied 70% to APEX site and then estimated the belowground biomass. We added the description in the text and cited the reference in this
revision.

*Page 8, line 173: Please correct date for late Holocene time frame*

We have corrected the date from "9 ka-1900AD" to "5ka-1900 AD".

*Page 9, lines 194-195: The Shuttle Radar Topography Mission (SRTM) only covered latitudes*
*56S-60N. Therefore there is NO SRTM data for Alaska. You obviously used some other source*
*for topography data – please provide correct reference.*

We directly used the data from He et al. (2014). We cited Zhuang et al. (2007) paper instead.

*Page 9, lines 197-203: Downscaling / bias correction is unclear. From the text one gets the*
*impression, that ECBILT fields and CRU data may be significantly different for 20th century.*
*However, my reading of the original publications is that bias correction minimised that*
*difference. Please clarify this – it would strongly strengthen the text.*

Yes, the climate data were bias-corrected to minimize the difference between ECBILT and CRU.
We made the clarification on this.

*Page 12, line 256: References to Figures 2 and 3 mixed up, please correct.*

We have made the change.

*Page 13, lines 279-289: Study sites unclear. Please provide table of site locations.*

Below is the table of descriptions of those sites. We cited Wang et al. (2016) in this revision

Sites used for comparison of carbon accumulation rates between simulation and observation [*Jones and*
*Yu,* 2010]

| Site name | Location | Peatland type | Latitude | Longitude | Dating method | No. of dates | Basal age (cal yr BP) | Time-weighted Holocene accumulation rates (g C m$^{-2}$ yr$^{-1}$) |
|---|---|---|---|---|---|---|---|---|
| Kenai Gasfield | Alaska, USA | fen | 60°27'N | 151°14'W | AMS | 12 | 11,408 | 13.1 |
| No Name Creek | Alaska, USA | fen | 60°38'N | 151°04'W | AMS | 11 | 11,526 | 12.3 |
| Horsetrail fen | Alaska, USA | rich fen | 60°25'N | 150°54'W | AMS | 10 | 13,614 | 10. 7 |

| Swanson fen | Alaska, USA | poor fen | 60º47'N | 150º49'W | AMS | 9 | 14,225 | 5.7 |

*Page 14, lines 314-316 and table 4: table 4 only lists uncertainty ranges for peatland vegetation. How were uncertainty ranges for upland vegetation derived? Certainly not from the ranges in table 4. Please clarify. Page 15, lines 333-334 – see comment #13.*

*Thankfully, the authors have used a spellchecker, so there are very few typos. However, there are numerous places in the text, where grammar needs checking: Temporal forms are not always consistent, and some sentences are missing single words or larger parts. Therefore CAREFUL COPY-EDITING is highly important.*

The uncertainties of the upland vegetation are from the uncertainties of parameters in previous study (Tang and Zhuang, 2008; 2009). We used the prior ranges of the parameters based on those studies. We added this statement in the text in this revision.  Following the referee's recommendation, we have carefully checked the whole text in terms of language.

**Response to Referee #2 (Dr. Tupek)**

We would like to thank Dr. Tupek for his thoughtful and constructive review, as well as his
detailed comments. Our responses to all of the referee's comments are provided as below.

*Authors tested SOC simulations from the modified version of their Terrestrial Ecosystem Model*
*(TEM) for peatlands (P-TEM) for Alaska region for time periods defined by climatic*
*characteristics (solar radiation, temperature, and precipitation levels) and vegetation*
*distribution during last 15 000 years. The model was applied for peatlands and non-peatlands*
*(mineral soil forests). Simulated C distributions, NPP, and peat depths were presented for*
*Alaska at fine scale spatial resolution (maps) and summarized for vegetation types. This can be*
*an interesting study if presented carefully*

Thanks for the constructive comments. Following your comments, we made all edits and
changes as detailed below. **The Method, Result, and Discussion sections were totally revised.**
**We also had revisions with respect to grammar and typos in the Abstract, Introduction and**
**Conclusions sections. However, those revisions were not marked. Please see the final**
**manuscript without tracks for a better vision.**

*The advances of the model include hydrology, soil thermal, C and N dynamics modules. This*
*looks promising however, the model code is not publicly available and need to be asked from*
*authors. No software details were given. I'd like to see the code and run an example simulation*
*in R to understand the model structure. Description of NPP simulation by the model is missing.*
*Description of Carbon and Nitrogen dynamics of CNDM module is also missing. Add*
*descriptions for clarity. Results largely depended on the adopted distribution of vegetation types.*
*Authors mentioned that distribution of vegetation types and their changes overdriven C*
*accumulation over climate, but also noted that climate had probably driven distribution of*
*vegetation types. Individual parameter values are not listed. It would be interesting to see the*
*changes of key parameters between vegetation types in relation to their prevailing climate.*

Response: The descriptions of P-TEM including modules of STM, HM, CNDM, and MDM were
less listed in this manuscript indeed. The main reason is that we have described the model
framework, specific parameters and the parameterization processes, along with the comparison
between simulation and observation regarding soil moisture, methane emissions, water-table
depth, and carbon and nitrogen dynamics in our previous study. Specifically, please find the
paper "**Quantifying Peat Carbon Accumulation in Alaska Using a Process-Based**
**Biogeochemistry Model**" for details (*Wang, S., Q. Zhuang, Z. Yu, S. Bridgham, and J. K.*
*Keller (2016), Quantifying peat carbon accumulation in Alaska using a process-based*
*biogeochemistry model, J. Geophys. Res. Biogeosci., 121, doi:10.1002/2016JG003452.*).

The above paper was under review by JGR-Biogeosciences at that time we submitted this
regional study to Biogeoscience. Now the paper was published, therefore we believe there is no
reason to repeat those model details. We have cited this newly published paper in the text in this
revision.

Our model source codes are written in C/C++, not in R. The model has a number of sub-models,
thus its source code is rather lengthy. We appreciate your interests in the model, we are open to
collaborate by using the model to do research so that we have sufficient time to understand the
code.

*Description of observations of peat depths that were used for model validation is not sufficient.*
*Some description is in section 2.5 but it is not clear and the points on the maps in Fig. 2 are*
*barely visible. Describe clearly.*

Response: The peat depths for model validation are from Hugelius et al. (2014). We used the
basal ages to determine the process of peatland expansion from Gorham et al. (2012). We added
our findings of the linkage between basal ages and vegetation distribution in the Result&
Discussion section (sec. 3.3) and discussed the related uncertainties. We also changed the symbol
size in Fig. 2 to make the figure of the basal age distribution much clearer to readers.

*Structure of the paper is unclear. Reorder the ideas, avoid using repetitions.*

Response: Thanks for the comments. In this revision, we re-ordered the ideas mainly in Result
and Discussion sections and made the structure more logical to readers. We also combined the
Result and Discussion sections to avoid repetitions. We shortened the Sec. 3.1, 3.2 and added the
text describing peat expansion and vegetation changes in Sec. 3.3.

*Methods are presented in results. Results are presented in discussion. For example lines 369-370*
*in results describe how peat depth was calculated for the first time. Discussion Section 4.3*
*presents too many numbers without deeper insights on reason behind differences between other*
*studies. At the end of discussion a scatterplot Fig. 14 between observed and modeled peat depths*
*is presented for the 1st time. The Fig. 14 shows that without exceptional agreement of 3 largest*
*values the rest of the scatter is just a gunshot indicating poor performance of the model in most*
*conditions. Authors avoid the explanation. Move results to result section. Present some values in*
*Tables. In discussion interpret the results with a focus on the model and data input.*

Response: Thanks for the comments, which helped us significantly improve our presentation of
the study.  In this revision, we combined the Result and Discussion sections. Also we have
shortened some sentences, only providing key points. We have moved the statement of the peat
depth calculation into the Method section. In the Result& Discussion section, we added some
contents regarding the reasons behind the vegetation changes through time, e.g., the relationships
between climate and vegetation distribution. Besides, we added some sentences to discuss the
uncertainties of the model and some assumptions in this study. This can represent the reasons
behind the differences between model simulation and observation. For the comparison of peat
depth, we used the peat sample at each site and directly compared the modeled depth with the
observation for that particular pixel. However, as the peat characteristics may differ from site to
site, even several kilometers apart, it is very hard for the model to really capture the true peat
depth. We added this statement at the end of the discussion. The model is suitable for being
applied at regional level and can capture the peat depths features (e.g., the mean values) in a
large area as showed similarity of the spatial peat depth distribution between our model and
observation (see Hugelius et al. (2014) Figure 3 for observed peat depth distribution).

*Interesting results as underestimation in uplands, lack of C loss simulation (Fig. 2), reasons*
*behind vegetation controlling C storage, disagreement with observations, assumption that*
*peatlands will remain C sink are brushed away. The agreement with other studies is OK but not*
*enough for discussion. Describe reasons for agreement/disagreements, give insights on*
*function/performance and reason why to use/trust your model. Although the authors claimed that*
*the PTEM includes CN module, nothing can be learned from reading the paper how this or other*
*modules affect the results. Given the SOC underestimation of uplands and large scatter with*
*peatlands, and large-scale climate estimates, could accounting for differences in nutrient status*
*or reevaluating response of C/N ratio be a key for improved estimation of spatial variability of*
*SOC accumulation of P-TEM or TEM model?*

Given the complexity of the biogeochemistry of peatlands, we were not able to address and
discuss all processes and mechanisms related to C and N in our simulations.  Rather we
dedicated our discussion on 1) how the features of vegetation distribution changes over time
affect carbon accumulation; 2) how different climates in various paleo-periods affect carbon
accumulation rates.  We admitted that we have not discussed how N affects C in the model.
However, P-TEM is a version that inherited all processes of C and N interactions in an early
version of TEM (Zhuang et al., 2003, McGuire et al., 1992), which has been extensively
evaluated and applied in numerous applications in northern high latitudes. Here we assume the C
and N interactions will also be valid for the peatland ecosystems.  In fact, we have started to use
C and N data collected from SPRUCE (Brain et al., 2016) experiment to evaluate the N
feedbacks to C dynamics. We expect to have a new version of P-TEM to fully account for C and
N feedbacks in peatland ecosystems.

*Authors claim that recent climate is warmer and wetter in summers and therefore with future*
*warmer-wetter climate peatland carbon sink will continue. Possibility of increased respiration*
*and C loss due to droughts or warmer winters is not mentioned. For the conclusion on future C*
*sink a simulations with climate scenarios would be useful.*

Agreed.  We have revised our statement regarding future carbon accumulation. We further stated
that further study is needed to have a better view on the soil carbon dynamic under future climate
scenarios. Given the complexity of peatland dynamics in terms of their areal changes (e.g.,
expansion and shrinkage, and new peatland formation), we feel analyzing how peatland carbon
responds to future climate changes and peatland dynamics is a tremendous work, which is well
beyond this study.  In addition, the current paper is already very long. Thus, we decided to
conduct a full analysis on Arctic peatland carbon responses to future climates in a separate study.

*Increase Figs. 2, 3, 4 and their legends size by 50 or 100%*

We have increased the legends size and symbol size to make the figures clearer.

Meanwhile, the sequence of some tables and figures has been adjusted to better present the
results.

*why are panels b, d, f in Fig. 3 scaled by zero? that makes differences to appear smaller increase*
*legend in Fig. 5*

We have decreased the axis scale in Fig. 3 to make the difference appear bigger. We also
increased the legend in Fig. 5.

*increase Fig. 7 and 9, why fig. 9 it has 2 legend bars?*

We have increased the size of each panel. We mistakenly put extra color bar in panel (f). We
have deleted it.

*line 137 "observed water contents drive STM"?, did you mean observed temperatures?*

The STM considers soil water content at different depths to simulate the soil temperature. We
used the observed soil water content at particular site to drive the STM when parameterizing the
model in *Wang, S., Q. Zhuang, Z. Yu, S. Bridgham, and J. K. Keller (2016), Quantifying peat*
*carbon accumulation in Alaska using a process-based biogeochemistry model, J. Geophys.*
*Res. Biogeosci., 121, doi:10.1002/2016JG003452.*. During the regional simulation, we used the
simulated soil water content at different depths directly from HM to drive the STM.

*lines 235-238 reformulate for clarity*

We have reformulated the sentences.

*line 293 correct value of C storage*

We have changed this value to 0.8 kg C m$^{-2}$.

*lines 300-302, 309 reformulate for clarity*

We have made the reformulation.

*line 315 range of what?*

The range here is for vegetation carbon storage. We have clarified this.

*lines 325 "spots were widely spread" reformulate, "SOC concentration" do you mean SOC*
*storage?*

We have reformulated them.

*line 326 reformulate "tundra was taking back area" or similar*

We have reformulated it.

*line 361 Table 4 is not showing parameters*

We have changed the statement to "due to uncertainties of observation" and cited the paper
which has the parameters listed.

*lines 375-377 reformulate for clarity*

We have reformulated the section.

*lines 420 – 460 reformulate for clarity*

We have reformulated the discussion part.

*line 424 why if p < 0.05 "some certain effects"?*

From the result in Wang et al. (2016) that p-value is less than 0.05, we can get the idea that the
interaction factor is one of the significant factors. We reorganized this part.

*line 428 "positive effect" of temperature? low temp slowed SOC accumulation, that's negative*
*effect*

Yes, it should be "negative effect" as the cooler condition during the neoglacier period slowed
the carbon accumulation. We reformulated this sentence.

*line 437 delete "suggesting the warmest climate during HTM" it comes by definition of HTM*

Correct. Repeating the definition of HTM can be awkward. We revised it.

*lines 443-448 does not make sense*

We deleted this part since it was talking about the uncertainties coming from the climate model
(ECBILT) itself, and has already been discussed in He et al. (2014). In this study, we directly
used the output of the model.

*line 452 what is "stored C in overall in the spatial scale"?*

We have reformulated the statement.

*line 454 "negative accumulation rate" avoid writing nonsense*

We have added "(meaning C loss)" right after this statement.

*lines 485 – 545 reformulate whole section, move results into results, check for repetitions,*
*highlight only most important trends and insights, shorten discussion if nothing much relevant to*
*say*

We have reformulated the result and discussion.

*lines 508-519 OK*

*References – change into the corresponding format of Biogeosciences*

Thanks for pointing this out. We have changed the reference and citation format according to
*Biogeosciences* submission requirement  The manuscript has never been submitted to other
journals before.

[revised manuscript text omitted]

---

## Referee Report (RR1)

Referee #2 Dr. Boris Ťupek, boris.tupek@luke.fi

Review of Wang et al. "Quantifying Soil Carbon Accumulation in Alaskan Terrestrial Ecosystems during the Last 15,000 Years"

General comments

Authors accounted for the required changes satisfactorily and the manuscript has improved. It is not clear if the long term variation in NPP is larger than the interannual NPP variation (Fig. 5). Please explain the reasons for the large NPP inter-annual variation. Is it annual variation in climate? One of the main findings is that vegetation distributions drives soil C. To me it seems that climate is driving vegetation distribution which determines soil C change. However, longterm vegetation distribution here is taken from maps produced for main periods of climatic change thus introducing large step wise changes. This is also interesting result. Consider reformulating.

Specific comments:

lines 32-34, reformulate, especially the origin of previous estimates is not obvious

lines 41-44 in abstract and lines 463-467 in conclusions are identical, reformulate or delete

lines 374-378 explain reasons for longterm variation and inter annual variation of NPP (Fig. 5).

Fig. 4 use same x axis; add a,b,c,d to the panels; Kenai gasfield mismatch?

Fig. 5 what is the reason for the large NPP inter-annual variation? add smoothed dashed line for higlighting the longterm changes ?

Fig.6 use same color codes as Fig. 5

Fig.7 use same color codes as Fig. 8?

Fig. 9 "the area of…0 km2" confusing/delete, divide SP and SBP peatlands?

Fig. 10 Peat C stock change. Specify that these are barplots to avoid confusion that peat C stock change is restricted to zero?

Fig. 13 use gray instead of cyan?

---

## Referee Report (RR2)

Review of "**Quantifying Soil Carbon Accumulation in Alaskan Terrestrial Ecosystems during the Last 15,000 Years**" by Sirui Wang, Qianlai Zhuang, and Zicheng Yu, revised version.

In their manuscript, the authors present a model study of soil carbon accumulation in Alaska over the last 15000 years with a special focus on peat carbon accumulation.

Compared to the authors' original submission I find the manuscript improved. However, a few issues remain.

This review of the revised version was, by the way, hindered by the fact that the "track changes" version of the manuscript changes was different from the submitted revision, therefore not showing the actual changes to the manuscript. I would also suggest to the authors to use the "compare documents" function the next time, since that shows differences between versions more clearly.

In my first review, one of my points was "Page 4, lines 94-95: the Spahni et al. Model has actually been evaluated with respect to the variables listed – see Wania et al. Publications on the LPJ-Why model on which Spahni is based." The authors have changed the passage I indicated to "In contrast, Spahni et al. (2013) used a dynamic global vegetation and land surface process model (LPX), based on LPJ (Sitch et al., 2003), imbedded with a peatland module, which considered the nitrogen feedback on plant productivity (Xu-Ri and Prentice, 2008) and plant biogeography, to simulate the SOC accumulation rates of northern peatlands. However, the model did not consider methane dynamics, which play an important role in affecting peat carbon dynamics, presumably due to its inadequate representation of ecosystem processes (Stocker et al., 2011, 2014; Kleinen et al., 2012). Furthermore, climatic effects on SOC were not fully explained."
Obviously the authors did not actually read the literature. The LPX model does indeed consider methane dynamics (Spahni et al., Biogeosciences, 2011 and Zürcher et al., Biogeosciences, 2013). In addition this statement is wrong in another way, since methane dynamics actually are not important at all in the peat carbon uptake, which is what the authors focus on in their manuscript. This passage needs revisiting. (Page 4, line 91 to page 5, line 97).
Furthermore, I asked the authors to provide a table with site locations used in their assessment at site level – instead they added a reference, which is inadequate. The aim of my request was to enable readers to quickly understand where these sites are, without requiring the original publications. In addition, the discussion of site reults is lacking some of the detail contained in the original manuscript.

I also asked the authors to describe how the change in peatland extent was determined. However, I was unfortunately not able to understand that from the description in the paper (page 9, lines 188-205). I understand the link between basal age and peatland extent the authors used to determine changes in peatland area, but that is very difficult to understand from the text since the connection is not made clearly. Please reformulate to make it clearer.

Page 13, line 277 refers to table 4, but this is table 2 in the revised version. I have not been able to check whether all other references to changed Figures and Tables are correct – I suggest the authors check this again before final publication.

---

## Author Response (AR2)

**Response to Referee #1 (Dr. Kleinen)**

We would like to thank Dr. Kleinen for his thoughtful and constructive review. Our responses to all of the referee's comments (italicized) are presented below.

*In their manuscript, the authors present a model study of soil carbon accumulation in Alaska over the last 15000 years with a special focus on peat carbon accumulation. Compared to the authors' original submission I find the manuscript improved. However, a few issues remain. This review of the revised version was, by the way, hindered by the fact that the "track changes" version of the manuscript changes was different from the submitted revision, therefore not showing the actual changes to the manuscript. I would also suggest to the authors to use the "compare documents" function the next time, since that shows differences between versions more clearly. In my first review, one of my points was "Page 4, lines 94-95: the Spahni et al. Model has actually been evaluated with respect to the variables listed – see Wania et al. Publications on the LPJ-Why model on which Spahni is based." The authors have changed the passage I indicated to "In contrast, Spahni et al. (2013) used a dynamic global vegetation and land surface process model (LPX), based on LPJ (Sitch et al., 2003), imbedded with a peatland module, which considered the nitrogen feedback on plant productivity (Xu-Ri and Prentice, 2008) and plant biogeography, to simulate the SOC accumulation rates of northern peatlands. However, the model did not consider methane dynamics, which play an important role in affecting peat carbon dynamics, presumably due to its inadequate representation of ecosystem processes (Stocker et al., 2011, 2014; Kleinen et al., 2012). Furthermore, climatic effects on SOC were not fully explained." Obviously the authors did not actually read the literature. The LPX model does indeed consider methane dynamics (Spahni et al., Biogeosciences, 2011 and Zürcher et al., Biogeosciences, 2013). In addition this statement is wrong in another way, since methane dynamics actually are not important at all in the peat carbon uptake, which is what the authors focus on in their manuscript. This passage needs revisiting. (Page 4, line 91 to page 5, line 97).*

We deleted those statements and only left the statement of "Climatic effects on SOC were not fully explained, presumably due to its inadequate representation of ecosystem processes".

*Furthermore, I asked the authors to provide a table with site locations used in their assessment at site level – instead they added a reference, which is inadequate. The aim of my request was to enable readers to quickly understand where these sites are, without requiring the original publications. In addition, the discussion of site results is lacking some of the detail contained in the original manuscript.*

In this revision, we added a table (Table 5) of the description of the four sites we used for site-level comparison. To make the manuscript more concise and focused, we decided not to discuss much on the site-level comparison in the Result and Discussion Section, since those results have already been presented and discussed in our previous study (Wang et al, 2016). Therefore, we
only showed the modeled results along with a brief discussion for those four sites in this study.

*I also asked the authors to describe how the change in peatland extent was determined. However,*
*I was unfortunately not able to understand that from the description in the paper (page 9, lines*
*188-205). I understand the link between basal age and peatland extent the authors used to*
*determine changes in peatland area, but that is very difficult to understand from the text since*
*the connection is not made clearly. Please reformulate to make it clearer.*

We have added few more sentences and reformulated to make such method clearer to readers.

*Page 13, line 277 refers to table 4, but this is table 2 in the revised version. I have not been able*
*to check whether all other references to changed Figures and Tables are correct – I suggest the*
*authors check this again before final publication.*

Thanks for pointing this out.  Correct, here we should have referred to Table 2 instead of Table 4.
In this revision, we checked all the references to figures and tables.

**Response to Referee #2 (Dr. Tupek)**

We would like to thank Dr. Tupek for his thoughtful and constructive review, as well as his detailed comments. Our responses to all of the referee's comments are provided as below.

*General comments:*

*Authors accounted for the required changes satisfactorily and the manuscript has improved. It is not clear if the long-term variation in NPP is larger than the inter-annual NPP variation (Fig. 5). Please explain the reasons for the large NPP inter-annual variation. Is it annual variation in climate? One of the main findings is that vegetation distributions drives soil C. To me it seems that climate is driving vegetation distribution which determines soil C change. However, long-term vegetation distribution here is taken from maps produced for main periods of climatic change thus introducing large step wise changes. This is also interesting result. Consider reformulating.*

Yes, the inter-annual NPP variation depends on the annual variation of climate. As key factors controlling plant productivity are monthly temperature, solar radiation, and precipitation, their inter-annual fluctuations affect NPP. However, despite the large inter-annual NPP variation, we can still see a clear trend of long-term increasing (or decreasing) NPP from 15 ka to 19[th]. The 1000-year average NPP of those several vegetation types mostly reached the highest during the HTM period. To make it clear for readers, we added a third panel in Fig. 5 to represent this long-term feature. The result of carbon dynamics indeed shows large step-wise changes due to vegetation distribution shift. We directly applied the vegetation maps, which were generated in previous study, and were discussed regarding the generating process of those maps along with the uncertainty analyses (He et al., 2014).

*Specific comments:*

*lines 32-34, reformulate, especially the origin of previous estimates is not obvious*

We added "using peat core data" to give a brief idea how the previous estimates have been done and the references and origin of the previous estimates were then discussed in Section 3.3.

*lines 41-44 in abstract and lines 463-467 in conclusions are identical, reformulate or delete*

In this revision, we revised the sentences in the Conclusion Section to avoid duplication.

*lines 374-378 explain reasons for long-term variation and inter annual variation of NPP (Fig. 5).*

We added the explanation of the reason causing inter-annual NPP variation here and made clearer that it was the trend of long-term NPP coincided with the warmer climate, higher vegetation C and soil C stocks during the HTM.

*Fig. 4 use same x axis; add a,b,c,d to the panels; Kenai Gasfield mismatch?*

We added those letters to each panel. We also used the same scale for x axis in each panel. We confirmed that there was no mismatch for Kenai Gasfield. The highest 500-years average rate occurred during 11-10.5 ka, as shown in the bar. We added "14.5-5 ka" in the caption to make
the bar clearer to read.

*Fig. 5 what is the reason for the large NPP inter-annual variation? add smoothed dashed line*
*for higlighting the longterm changes?*

We added the reason in the text and added a separate panel (Figure 5c) to show the long-term
changes of NPP.

*Fig.6 use same color codes as Fig. 5*

We applied the same color code to Fig. 6.

*Fig.7 use same color codes as Fig. 8?*

We used the same color code in Fig. 7.

*Fig. 9 "the area of…0 km2" confusing/delete, divide SP and SBP peatlands?*

We deleted this confusing sentence.

*Fig. 10 Peat C stock change. Specify that these are barplots to avoid confusion that peat C stock*
*change is restricted to zero?*

We specified that the plots are bars.

[revised manuscript text omitted]